# SD-MoE: Spectral Decomposition for Effective Expert Specialization

Ruijun Huang [* 1 2]  Fang Dong [* 1]  Xin Zhang [* 1]  Anrui Chen [1]  Hengjie Cao [1 3]  Zhendong Huang [1]  Jixian Zhou [1]
Mengyi Chen [1]  Yifeng Yang [1]  Mingzhi Dong [4]  Yujiang Wang [5 6]  Jinlong Hou [3]  Qin Lv [7]  Robert P. Dick [8]
Yuan Cheng [3]  Fan Yang [1]  Tun Lu [1]  Chun Zhang [9]  Li Shang [1 10]

## Abstract

Mixture-of-Experts (MoE) architectures scale Large Language Models via expert specialization induced by conditional computation. In practice, however, expert specialization often fails: some experts become functionally similar, while others functioning as de facto shared experts, limiting the effective capacity and model performance. In this work, we analyze expert specialization from a spectral perspective on parameter and gradient spaces, uncovering that (1) experts share highly overlapping dominant spectral components in their parameters, (2) dominant gradient subspaces are strongly aligned across experts, driven by ubiquitous low-rank structure in human corpus, and (3) gating mechanisms preferentially route inputs along these dominant directions, further limiting specialization. To address this, we propose Spectral-Decoupled MoE (SD-MoE), which decomposes both parameter and gradient in the spectral space. SD-MoE improves performance across downstream tasks, enables effective expert specialization, incurring minimal additional computation, and can be seamlessly integrated into a wide range of existing MoE architectures, including Qwen and DeepSeek.

## 1. Introduction

Mixture-of-Experts (MoE) scales Large Language Models (LLM) via expert specialization induced by conditional computation. In practice, however, specialization in LLM MoEs often degrades: some experts are nearly interchangeable with minimal performance impact (Zheng et al., 2025; Zhang et al., 2025), while others are activated almost universally, effectively acting as shared experts (Cai et al., 2025; Guo et al., 2026). Despite its prevalence, the underlying causes of this failure of specialization remain poorly understood.

This paper addresses this gap through a spectral analysis of both parameter and gradient spaces. We formalize expert parameter overlap, uncover the training dynamics that give rise to limited specialization, and propose a new MoE architecture that enables effective specialization among experts. Our key findings are summarized as follows:

**Experts share highly aligned dominant spectral components, even when a common expert is explicitly used.** As shown in Figure 1, a large fraction of spectral energy is concentrated in a small number of leading singular directions, and these dominant subspaces are highly aligned across experts with a similarity larger than 0.9. Even in models that explicitly employ a shared (or "common") expert, such as Qwen (Qwen-Team, 2024) and DeepSeek (Dai et al., 2024), the weight matrices of unique experts still exhibit strongly overlapping dominant spectral directions.

**Dominant gradient subspaces are highly aligned across experts, driving parameter overlap.** As illustrated in Figure 2(a), the dominant low-rank components of expert gradients exhibit strong cross-expert alignment, reflecting a shared update direction during training. In contrast, the remaining long-tail directions exhibit substantially lower cross-expert similarity, suggesting that they capture more expert-specific variations. As detailed in Section 2.2, this alignment arises from a shared low-rank structure in input representations, which induces a common right-singular subspace in expert gradients and consequently aligns their dominant parameter updates.

**Gating is dominated by leading spectral structure, result-**

---

[*]Equal contribution  [1]Fudan University, Shanghai, China
[2]Greater Bay Area National Center of Technology Innovation, Research Institute of Tsinghua University in Shenzhen, Shenzhen, China [3]Shanghai Innovation Institute, Shanghai, China [4]University of Bath, Bath, United Kingdom [5]Department of Engineering Science, University of Oxford, Oxford, UK [6]Oxford Suzhou Centre for Advanced Research, University of Oxford, Suzhou, China [7]Department of Computer Science, University of Colorado Boulder, Colorado, USA [8]Department of Electrical Engineering and Computer Science, University of Michigan [9]Research Institute of Tsinghua University in Shenzhen, Shenzhen, China [10]Shenzhen Loop Area Institute, Shenzhen, China. Correspondence to: Li Shang <lishang@fudan.edu.cn>.

*Proceedings of the $43^{rd}$ International Conference on Machine Learning*, Seoul, South Korea. PMLR 306, 2026. Copyright 2026 by the author(s).

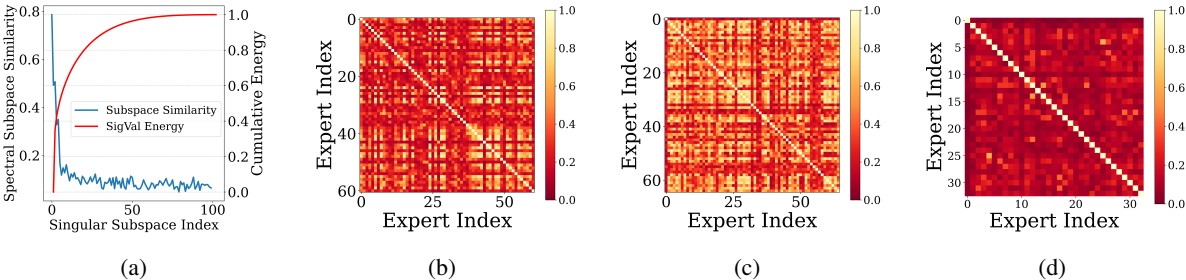

*Figure 1.* **(a)** The top 1% spectral subspace in DeepSeek model is highly aligned (similarity 0.8) and carries >30% of the energy (Ethayarajh, 2019; Puccetti et al., 2022; Cao et al., 2026), while the remaining 99% tail is weakly aligned (~0.1). **(b–d)** Pairwise spectral similarity of expert parameters for **(b)** Qwen1.5-MoE (Qwen-Team, 2024), **(c)** DeepSeek-V2-Light (Dai et al., 2024), and **(d)** SD-MoE. Existing MoE models exhibit strong overlap in the top 1% spectral subspace (avg. 0.7; some >0.9), whereas SD-MoE reduces this similarity to ~0.1. Supplementary results in more models and layers are in Appendix A Figure 11-13.

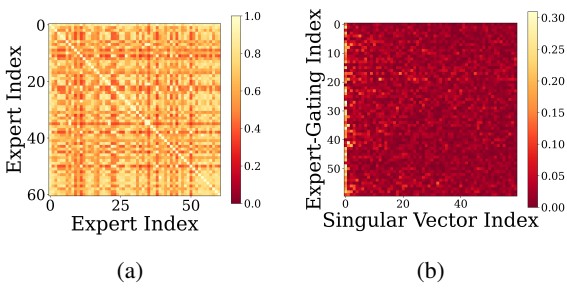

*Figure 2.* Analysis of Qwen1.5-MoE-A2.7B. **(a)** Pair-wise principal similarity of the dominant low-rank subspace from the gradient matrices of all experts. High values indicate near-identical spectral directions across experts, suggesting gradient similarity in their low-rank subspace. **(b)** The row vectors of the gating matrix exhibit high alignment with the leading singular directions of the expert weight matrices, indicating that the gating mechanism is dominated by common information.

**ing in non-specialized routing.** As shown in Figure 2(b), the row vectors of gating weight matrices exhibit strong alignment with the dominant spectral directions of expert weight matrices. This indicates that the gating mechanism itself fails to fulfill its intended role of promoting expert specialization; instead, it routes inputs based on the same shared low-rank features, which undermines effective expert specialization.

These findings reveal that shared dominant spectral components induce parameter redundancy and gating bias, diminishing the effective capacity of existing MoE models and ultimately degrading downstream performance (Zhou et al., 2022; Qiu et al., 2025; Guo et al., 2026).

To address this spectral entanglement, we propose *Spectral-Decoupled MoE (SD-MoE)*. SD-MoE decomposes each expert matrix into a shared low-rank spectral subspace, and an orthogonal, expert-specific complement in the long-tail subspace. During training, the gradient is similarly decom-

posed: the projection onto the shared subspace updates the common expert, while the residual updates corresponding unique experts. SD-MoE provides three key advantages:

• *Improved performance:* The method yields on average 3% gains on downstream tasks and 30% training efficiency improvement.

• *Effective expert specialization:* SD-MoE reduces inter-expert similarity to below 0.1 (Figure 1(d)) and enabling more diverse semantic routing.

• *Low overhead and broad applicability:* SD-MoE introduces only 5% additional computational cost and can be integrated seamlessly into existing MoE architectures, including Qwen (Qwen-Team, 2024) and DeepSeek (Dai et al., 2024).

## 2. Spectral Analysis for Expert Specialization

This section analyzes expert specialization failure through three lenses: parameter redundancy, gradient alignment, and gating bias in a shared dominant spectral subspace.

Our analysis centers on widely used open-source MoE models Qwen (Qwen-Team, 2024) and DeepSeek (Dai et al., 2024), which replace the standard FFN with $E$ routed experts. For each token representation $\mathbf{x}_t \in \mathbb{R}^d$, a router selects a small subset of experts $\mathcal{E}_t \subseteq \{1, \ldots, E\}$ and aggregates their outputs as $\mathbf{h}_t = \sum_{i \in \mathcal{E}_t} g_{t,i} \operatorname{FFN}_i(\mathbf{x}_t)$, where $g_{t,i} \geq 0$ are weights over $i \in \mathcal{E}_t$ (typically normalized to sum to 1). Each expert is a SwiGLU block: $\operatorname{FFN}_i(\mathbf{x}) = \mathbf{W}_D^{(i)}(\sigma(\mathbf{W}_G^{(i)}\mathbf{x}) \odot (\mathbf{W}_U^{(i)}\mathbf{x}))$, where $\odot$ is element-wise product.

### 2.1. Spectral Overlap in Expert Parameters

*Objective 1: Character expert specialization in parameter spectral space.* We begin by examining specialization at the level of expert parameters. In a MoE architecture, spe-

cialization implies that different experts implement *distinct transformations* (Guo et al., 2026). We therefore characterize each expert matrix through singular value decomposition (SVD): its singular vectors identify the input and output directions on which the corresponding linear transformation acts, while its singular values quantify the strength of these directions. This makes SVD a natural tool for comparing whether experts share dominant functional directions.

*Dominant spectral subspaces and overlap metric.* Formally, consider a linear projection in expert $i$ and write its transformation as $\mathbf{y} = \mathbf{W}^{(i)}\mathbf{x}$, where $\mathbf{W}^{(i)} \in \mathbb{R}^{d_{\mathrm{out}} \times d_{\mathrm{in}}}$, $\mathbf{x} \in \mathbb{R}^{d_{\mathrm{in}}}$, and $\mathbf{y} \in \mathbb{R}^{d_{\mathrm{out}}}$. With singular value decomposition,

$$\mathbf{W}^{(i)} = \mathbf{U}^{(i)}\mathbf{\Sigma}^{(i)}\mathbf{V}^{(i)\top}, \tag{1}$$

the columns of $\mathbf{V}^{(i)}$ form orthonormal directions in the input feature space, while the columns of $\mathbf{U}^{(i)}$ form orthonormal directions in the output feature space.

SVD provides singular directions for both the input and output spaces, so we must choose the appropriate basis for comparing experts. In SwiGLU, the up- and gate-projections operate in the *input* feature space, whereas the down-projection mixes directions in the *output* feature space. Accordingly, we define the comparison basis for each expert matrix as

$$\mathbf{B}^{(i)} := \begin{cases} \mathbf{V}^{(i)}, & \text{up\_proj / gate\_proj (input feature space)}, \\ \mathbf{U}^{(i)}, & \text{down\_proj (output feature space)}. \end{cases}$$

For any spectral index interval $[s : e]$, we define the corresponding subspace of expert $i$ as

$$\mathcal{S}^{(i)}_{[s:e]} := \mathrm{span}\left(\mathbf{B}^{(i)}_{s:e}\right),$$

where $\mathbf{B}^{(i)}_{s:e}$ denotes columns $s, \ldots, e$ of $\mathbf{B}^{(i)}$.

To quantify specialization, we measure the overlap between expert subspaces using principal subspace similarity:

$$\mathrm{sim}\left(\mathcal{S}^{(i)}_{[s:e]}, \mathcal{S}^{(j)}_{[s:e]}\right) := \lambda_{\max}\left(\mathbf{B}^{(j)\top}_{s:e}\mathbf{B}^{(i)}_{s:e}\right), \tag{2}$$

where $\lambda_{\max}(\cdot)$ denotes the largest singular value. Higher values indicate stronger alignment between expert subspaces and thus weaker specialization.

*Finding 1: Dominant spectral subspaces are highly aligned across expert parameters.* Applying Eq. equation 2 across consecutive $1\%$ spectral intervals (and reporting the corresponding energy CDF), we observe a sharp head–tail contrast in inter-expert similarity (Fig. 1): the first $1\%$ interval attains very high similarity $\approx 0.8$, whereas the remaining intervals are much lower, roughly $\approx 0.1$ on average. Together with the strongly concave energy CDF, this suggests that a small leading subspace both carries most of the transformation energy and dominates inter-expert similarity. Therefore,

in the rest of this paper we refer to the dominant-subspace as the top $1\%$ directions unless otherwise specified.

*Objective 2: Analyze the functional role of shared dominant spectral subspaces.* The overlap metric captures similarity in expert parameters, but does not reveal the *functional role* of the shared dominant subspace. To understand what these shared parameter directions represent, we further analyze how they are activated by real language inputs.

For any input token $\mathbf{x} \in \mathbb{R}^{d_{\mathrm{in}}}$, its projection onto the $m$-th singular direction is $\langle \mathbf{v}_m, \mathbf{x} \rangle$. We say that $\mathbf{x}$ *activates* the $m$-th direction if $|\langle \mathbf{v}_m, \mathbf{x} \rangle| > \tau$, where $\tau$ is a layer-specific threshold chosen to match the scale of activations in each layer. All activation experiments are conducted using DCLM data (Li et al., 2024) on Qwen1.5-MoE (Qwen-Team, 2024).

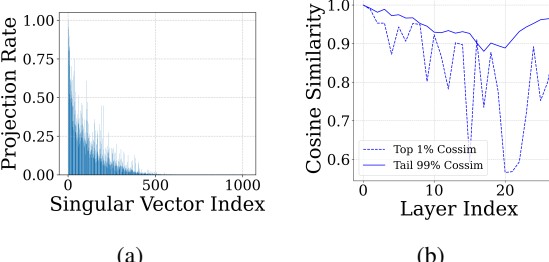

(a)  (b)

*Figure 3.* **(a)** Activation ratio of each singular direction, defined as the fraction of tokens whose projections exceed the activation threshold. Leading singular directions are activated by nearly all tokens, indicating that they encode features consistently present across tokens and domains. **(b)** After shuffling input tokens, the activation projections onto the top $1\%$ singular directions changes larger than that for the tail $99\%$, indicating that the dominant subspace is highly sensitive to syntactic structure.

As shown in Figure 3(a), the activation ratios across singular directions exhibit a pronounced long-tailed distribution. The leading singular directions are activated by nearly all input tokens, whereas the vast majority of directions in the spectral tail are rarely activated. This indicates that the dominant subspace encodes broadly applicable features shared across inputs, rather than expert-specific behavior.

*Finding 2: Dominant spectral subspace in parameters encodes mainly syntactic structures.* To further characterize the information encoded by the dominant subspace, we shuffle input tokens to see how projection varies. As shown in Figure 3(b), we compute the cosine similarity between activations projected onto the top $1\%$ and tail $99\%$ singular directions. Shuffling causes a larger similarity drop for the dominant subspace, indicating that these directions are highly sensitive to syntactic structure. This suggests that the shared dominant parameter directions encode general-purpose linguistic knowledge, such as syntax, rather than expert-specialized content.

## 2.2. Shared Spectral Subspace in Expert Gradients

*Objective 3: Identify the training dynamics responsible for parameter overlap across experts.* Having established that expert parameters exhibit highly aligned dominant subspaces, we now turn to the *training dynamics* that give rise to this phenomenon. We analyze the spectral structure of expert gradients to understand how such alignment emerges across experts.

*Spectral geometry of expert gradients.* Consider a same linear projection in expert $i$ $\mathbf{y} = \mathbf{W}^{(i)}\mathbf{x}$, (with $\mathbf{W}^{(i)} \in \mathbb{R}^{d_{\text{out}} \times d_{\text{in}}}$) as defined in Section 2.1, and let $\mathcal{B}_i$ denote the routed mini-batch for expert $i$ at a given training step. The gradient of the loss with respect to $\mathbf{W}^{(i)}$ is

$$\mathbf{G}^{(i)} := \frac{\partial \mathcal{L}(\mathcal{B}_i)}{\partial \mathbf{W}^{(i)}} = \sum_{t \in \mathcal{B}_i} \frac{\partial \ell_t}{\partial \mathbf{y}_t} \frac{\partial \mathbf{y}_t}{\partial \mathbf{W}^{(i)}} = \sum_{t \in \mathcal{B}_i} \boldsymbol{\delta}_t \mathbf{x}_t^\top, \quad (3)$$

where $\mathbf{x}_t \in \mathbb{R}^{d_{\text{in}}}$ is the input vector of token $t$, $\mathbf{y}_t = \mathbf{W}^{(i)}\mathbf{x}_t$ is the token-level output of this linear map on token $t$, and $\boldsymbol{\delta}_t := \frac{\partial \ell_t}{\partial \mathbf{y}_t} \in \mathbb{R}^{d_{\text{out}}}$ is the upstream error signal. The singular value decomposition (SVD) of the gradient is thus $\mathbf{G}^{(i)} = \mathbf{U}_G^{(i)} \mathbf{\Sigma}_G^{(i)} (\mathbf{V}_G^{(i)})^\top$.

For concreteness, we focus on the input-side subspace and use the right singular vectors $\mathbf{V}_G^{(i)}$ in the following analysis. The output-side case is completely analogous as is discussed in Section 2.1. We therefore define the dominant gradient-update subspace as

$$\mathcal{S}_G^{(i)} := \text{span}\left(\mathbf{V}_{G,1:k}^{(i)}\right), \qquad \mathbf{V}_{G,1:k}^{(i)} := \mathbf{V}_G^{(i)}[:, 1\!:\!k],$$

with $k$ chosen to capture the top $1\%$ of singular directions (matching our parameter analysis). Similar to Equation 2, cross-expert gradient alignment is also quantified via principal subspace similarity:

$$\text{sim}\left(\mathcal{S}_G^{(i)}, \mathcal{S}_G^{(j)}\right) := \lambda_{\max}\left((\mathbf{V}_{G,1:k}^{(j)})^\top \mathbf{V}_{G,1:k}^{(i)}\right). \quad (4)$$

*Finding 3: Dominant gradient subspaces are highly aligned across experts.* Figure 4 reveals a striking pattern: the dominant $1\%$ gradient subspaces exhibit high alignment across all expert pairs (Figure 4(a)), despite receiving disjoint sets of tokens through gating. In contrast, the long-tail $99\%$ subspace shows significantly weak alignment (Figure 4(b)). This indicates that the *primary direction of parameter updates* is effectively shared among all experts.

*Objective 4: Explain the origin of cross-expert alignment in dominant gradient subspaces.* Motivated by Finding 3, we seek to explain why the input-side dominant gradient-update subspace $\mathcal{S}_G^{(i)} = \text{span}(\mathbf{V}_{G,1:k}^{(i)})$ remains highly similar across experts despite receiving disjoint routed tokens.

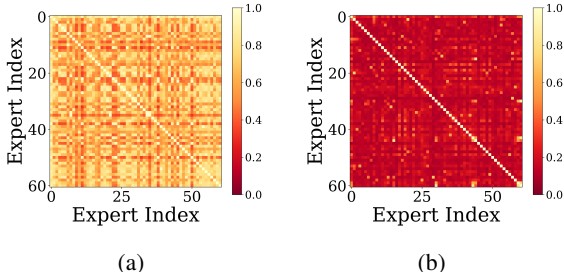

(a)            (b)

*Figure 4.* Gradient spectral analysis of experts in Qwen1.5-MoE-A2.7B. Pairwise principal similarity of **(a)** the dominant low-rank ($1\%$) gradient subspace and **(b)** the remaining long-tail subspace across all experts, including the common expert. High similarity in the low-rank subspace indicates shared gradient directions across experts, while tail similarity is substantially weaker. Additional results across models and layers are provided in Appendix A (Figures 14 and 15).

We investigate a representation-driven explanation: whether input activations exhibit a shared dominant subspace $C$ that persists across diverse samples and routed mini-batches, and whether this shared structure can induce a common component in each expert gradient through $\frac{\partial \ell}{\partial \mathbf{W}} = \boldsymbol{\delta} \mathbf{x}^\top$.

*Finding 4: Shared low-rank structure in input representations drives gradient alignment.* To validate our hypothesis, we compute activation tensors (shaped [sequence length, dimension]) from randomly sampled sequences and compute the principal similarity in their top $1\%$ spectral subspace. As shown in Figure 5, these dominant subspaces exhibit high pairwise cosine similarity across diverse data samples. This confirms the existence of a shared low-rank structure in the input representation space.

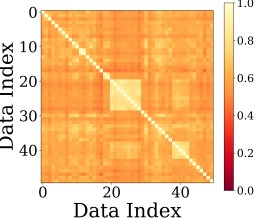

*Figure 5.* Pairwise principal subspace similarity of the top $1\%$ activation spectral directions across data samples, confirming the presence of a shared low-rank input subspace. Additional results across models and layers are provided in Appendix A (Figure 16).

We then outline the mathematical mechanism by which shared input structure drives gradient alignment. For the same linear projection $\mathbf{y} = \mathbf{W}^{(i)}\mathbf{x}$, the per-token gradient is $\frac{\partial \ell}{\partial \mathbf{W}} = \boldsymbol{\delta} \mathbf{x}^\top$, which is similar to Equation 3. Let $\mathbf{P}_C$ denote the orthogonal projector onto $C$, and define $\mathbf{P}_{C^\perp} := \mathbf{I} - \mathbf{P}_C$. Decomposing each input into a shared dominant subspace and a long-tailed subspace $\mathbf{x}_t = \mathbf{P}_C \mathbf{x}_t + \mathbf{P}_{C^\perp} \mathbf{x}_t$,

the batch gradient becomes

$$\mathbf{G}^{(i)} = \underbrace{\sum_{t \in \mathcal{B}_i} \boldsymbol{\delta}_t (\mathbf{P}_C \mathbf{x}_t)^\top}_{\mathbf{G}_C^{(i)}} + \underbrace{\sum_{t \in \mathcal{B}_i} \boldsymbol{\delta}_t (\mathbf{P}_{C^\perp} \mathbf{x}_t)^\top}_{\mathbf{R}^{(i)}} . \quad (5)$$

The term $\mathbf{G}_C^{(i)}$ presented in *every* expert's update acts entirely within the shared subspace $C$. When $C$ captures most of the input energy (as observed), $\mathbf{G}_C^{(i)}$ dominates the spectral norm of $\mathbf{G}^{(i)}$, forcing the leading right singular vectors of $\mathbf{G}^{(i)}$ to concentrate near $C$.

We observe the same phenomenon under a different optimizer on Moonlight-16B-A3B, a Muon-trained MoE model, with detailed results provided in Appendix G.

### 2.3. Spectral Bias in Gating Decisions

*Objective 5: Examine the impact of spectral dominance on gating behavior.* Having shown that both activations and expert parameters have a shared dominant subspace, we now examine how the *gating mechanism* is influenced by this spectral dominance.

*Gating as directional matching.* In standard Mixture-of-Experts (MoE) layers, the router computes a distribution over experts via

$$\mathbf{s} = \mathrm{softmax}(\mathbf{W}_{gate}\mathbf{x})$$

, where $\mathbf{x} \in \mathbb{R}^{d_{\mathrm{in}}}$ is the input token representation and $\mathbf{W}_{gate} \in \mathbb{R}^{E \times d_{\mathrm{in}}}$ is a learnable gating matrix, where $E$ is the number of experts. The $i$-th gating score is therefore

$$s_i \propto \exp(\mathbf{w}_i \mathbf{x})$$

, where $\mathbf{w}_i$ denotes the $i$-th *row* of $\mathbf{W}_{gate}$. This reveals that gating is fundamentally a process of *directional matching*: each expert is associated with a direction $\mathbf{w}_i$ in the input space, and tokens are assigned based on their inner product with these expert-specific vectors.

Consequently, we examine how these gating directions $\{\mathbf{w}_i\}_{i=1}^E$ relate to the spectral subspaces.

*Finding 5: Gating aligns with shared spectral subspaces, fails to specialize.* We evaluate, for each $i$, the alignment between the gating vector $\mathbf{w}_i$ and the dominant subspaces of (a) the input activations and (b) the parameters of the corresponding expert. Similar to principal similarity defined in Equation 2 As shown in Figure 6(a), gating vectors exhibit strong alignment with the leading singular directions of the activation matrix. Moreover, Figure 6(b) reveals consistent alignment between each gating vector $\mathbf{w}_i$ and the leading singular vector of its associated expert's parameter matrix.

Taken together, the parameter, gradient, and gating analyses reveal a mechanistic link behind expert under-specialization.

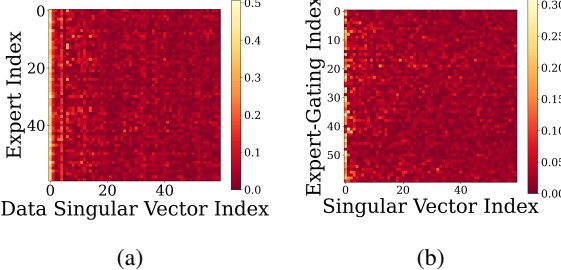

(a)            (b)

*Figure 6.* Gate matrix analysis in Qwen1.5-MoE (Qwen-Team, 2024). The alignment of row vectors of the gating matrix with **(a)** the leading spectral directions of the data activation **(b)** the leading singular directions of the expert parameters. Gating mechanism is dominated by common information. Supplementary results in more models and layers are in Appendix A Figure 17 and 18.

Shared low-rank structure in language representations induces aligned dominant gradient components across experts, which accumulate into overlapping dominant parameter subspaces. The router then compares experts mainly through these shared directions rather than expert-specific ones, making experts less distinguishable to the gate and weakening routing selectivity.

## 3. Spectral-Decoupled MoE

This section introduces Spectral-Decoupled Mixture of Experts (SD-MoE), a spectral decomposition-based approach that directly addresses the considerable overlap among experts in parameter and gradient spaces. As demonstrated in Section 4, this design reduces expert overlap by 70%, improves training efficiency by 30% on MoE architectures based on DeepSeek and Qwen, indicating its broad applicability across diverse existing MoE architectures.

### 3.1. Expert Spectral Decomposition

SD-MoE optimizes expert specialization though (i) spectrally decouples each expert's parameter matrix into a shared low-rank component and an expert-specific component within its orthogonal complement; and (ii) decouples the gradient updates for these two components to enable independent optimization. This design reduces inter-expert gradient interference and variance, relaxes learning rate constraints, and accelerates convergence.

Figure 7 uses a two-layer MLP as an example to illustrate how to apply SD-MoE to the standard linear-gate MoE paradigm. Given an input token $\mathbf{x}$, routing scores are computed as $\mathbf{W}_{\mathrm{gate}}\mathbf{x}$, where $\mathbf{W}_{\mathrm{gate}}$ is a learnable gating matrix. The top-n experts with the highest scores are selected for activation. Each expert $i$ consists of two weight matrix: an up-projection $\mathbf{W}_U^{(i)}$ and a down-projection $\mathbf{W}_D^{(i)}$. The

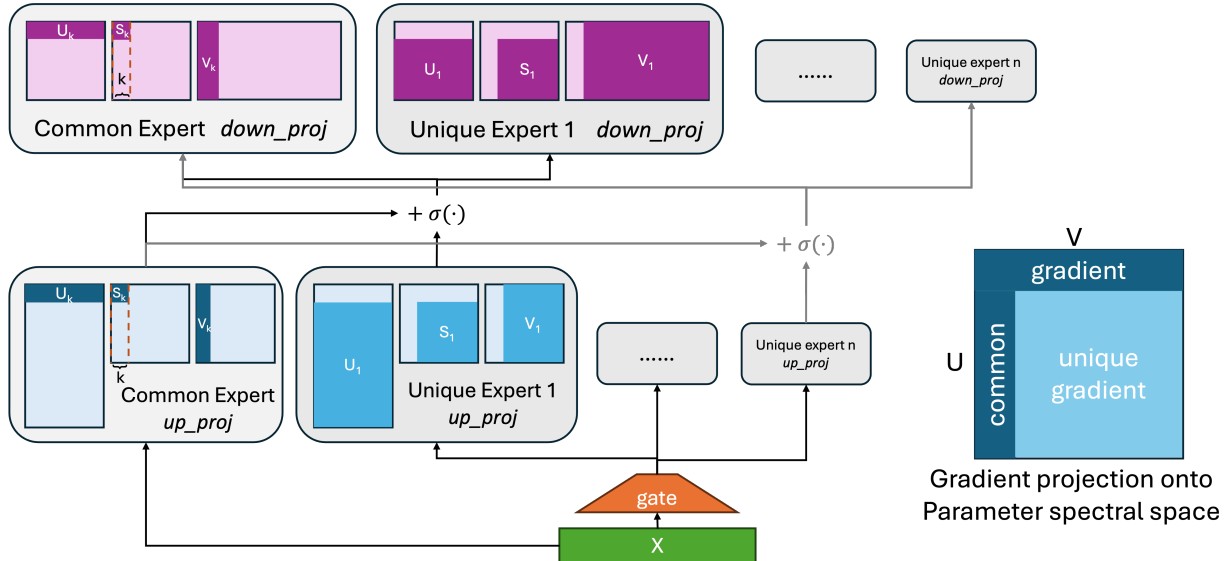

*Figure 7.* The architecture of SD-MoE. SD-MoE decompose each linear matrix into a shared low-rank subspace and multiple unique components in the orthogonal complement. During gradient updates, the gradient is also decomposed into these subspaces to update the experts accordingly.

output is computed as $\mathbf{W}_D^{(i)}\sigma(\mathbf{W}_U^{(i)}\mathbf{x})$, where $\sigma(\cdot)$ denotes the activation function.

However, architectural decoupling alone is insufficient to ensure sustained specialization during training. Section 3.2 addresses this by spectrally decoupling expert parameters at initialization, preventing dominant shared components from biasing early optimization, while Section 3.3 further decomposes gradients during training to prevent cross-expert interference from re-emerging. Together, these steps ensure that spectral decoupling is preserved throughout training.

### 3.2. Parameter spectral decoupling at initialization

For each weight matrix in the experts, SD-MoE decomposes it into (i) a shared component $\mathbf{W}_c$ for the common expert, and (ii) multiple expert-specific components $\mathbf{W}_u^{(i)}$ for unique experts. Let $\mathbf{U}\mathbf{\Sigma}\mathbf{V}^\top$ be random sampled singular vectors and singular values, $k$ be the rank of dominant subspace. We define $\mathbf{U}_k = \mathbf{U}_{[:,:k]}$, $\mathbf{V}_k = \mathbf{V}_{[:,:k]}$, $\mathbf{\Sigma}_k = \mathbf{\Sigma}_{[:k,:k]}$. The matrix in the common expert is initialized as

$$\mathbf{W}_c := \mathbf{U}_k\,\mathbf{\Sigma}_k\,\mathbf{V}_k^\top, \tag{6}$$

The matrix in each unique expert $\mathbf{W}_u^{(i)}$ is then initialized such that it lies in the orthogonal complement of the common subspace:

$$\mathbf{U}_k^\top\mathbf{W}_u^{(i)} = \mathbf{0}, \quad \mathbf{W}_u^{(i)}\mathbf{V}_k = \mathbf{0}. \tag{7}$$

Details of how unique experts are initialized are deferred to Appendix C.

### 3.3. Gradient spectral decomposition during training

Before detailing our gradient decomposition strategy, we first outline the forward pass, which determines how gradients are subsequently computed.

**Forward Pass** The routing mechanism in SD-MoE follows the standard linear MoE router. If a token $\mathbf{x}$ is routed to expert $i$, SD-MoE computes the forward pass of each linear matrix in the expert as

$$\mathbf{W}^{(i)}\mathbf{x} = (\mathbf{W}_c + \mathbf{W}_u^{(i)})\mathbf{x}. \tag{8}$$

where we refer to $\mathbf{W}^{(i)}$ as a *proxy expert matrix* corresponding to the unique expert $\mathbf{W}_u^{(i)}$. The same operation is applied to all linear matrices, and thus the output is $\mathbf{W}_D^{(i)}\sigma(\mathbf{W}_U^{(i)}\mathbf{x})$ where $\mathbf{W}_U^{(i)} = \mathbf{W}_{U,c} + \mathbf{W}_{U,u}^{(i)}$, $\mathbf{W}_D^{(i)} = \mathbf{W}_{D,c} + \mathbf{W}_{D,u}^{(i)}$.

**Gradient Decomposition** During back-propagation, each proxy expert matrix $\mathbf{W}^{(i)}$ produces a gradient $\mathbf{G}^{(i)} = \frac{\partial \mathcal{L}}{\partial \mathbf{W}^{(i)}}$. We decompose this gradient into two parts: (i) a *low-rank* component $\mathbf{G}_c^{(i)}$ used to update the common expert matrix $\mathbf{W}_c$, and (ii) a *long-tail* component $\mathbf{G}_u^{(i)}$ used to update the unique expert matrix $\mathbf{W}_u^{(i)}$. Let $\mathbf{P}_U = \mathbf{U}_k\mathbf{U}_k^\top$ and $\mathbf{P}_V = \mathbf{V}_k\mathbf{V}_k^\top$. The gradient is decomposed as

$$\mathbf{G}_c^{(i)} := \mathbf{P}_U\mathbf{G}^{(i)} + (\mathbf{I} - \mathbf{P}_U)\,\mathbf{G}^{(i)}\,\mathbf{P}_V, \tag{9}$$

$$\mathbf{G}_u^{(i)} := \mathbf{G}^{(i)} - \mathbf{G}_c^{(i)} \tag{10}$$

An intuitive illustration of this decomposition is shown in Figure 7. In the spectral space spanned by the left and right

singular subspaces $\mathbf{U}$ and $\mathbf{V}$, gradient components involving the dominant low-rank subspaces $\mathbf{U}_k \mathbf{V}_k^\top$ are absorbed into $\mathbf{G}_c^{(i)}$, while the remaining components in the double orthogonal complement are assigned to $\mathbf{G}_u^{(i)}$. Since the basis for the common expert may change after update, we perform SVD on $\mathbf{W}_c$ periodically to obtain correct $\mathbf{U}_k, \mathbf{V}_k$.

# 4. Experiments

## 4.1. Experiment Setup

Our experiments are based on widely available open-source MoEs DeepSeek (Dai et al., 2024) and Qwen (Yang et al., 2025), which are explicitly designed to encourage expert specialization. We conduct comparisons at two scales: (2B/0.8B) and (7B/1.5B). All models are trained on a 100B subset of the DCLM corpus (Li et al., 2024). The shared expert subspace rank is set to 8, and we empirically fix the periodic SVD update interval to every 16 steps. Full details on hyperparameters are provided in Appendix D.

## 4.2. Main Results

On downstream tasks (Table 1), SD-MoE delivers better performance across nearly all benchmarks compared to baselines. Notably, spectral sharing and decomposition further enable training with larger learning rates with out instability, delivering a 30% training efficiency. As is shown in Figure 8, while baseline Qwen model diverges at learning rates beyond $1 \times 10^{-4}$, SD-MoE remains stable up to a 4× increase $4 \times 10^{-4}$. More details for the larger learning rate experiments can be referred to in Appendix E.

We also provide additional evaluations in Appendix H.1, including comparisons with other specialization baselines, long-text tasks, training-dynamics curves, and an effective-capacity comparison. These results further show that SD-MoE consistently improves over strong MoE baselines beyond the final checkpoint and standard multiple-choice evaluations.

**Spectral conditioning and training stability.** Prior work has shown that training stability and convergence speed are closely tied to the spectral geometry of network mappings: well-conditioned singular spectra, often described through *dynamical isometry*, permit larger stable learning rates and faster optimization (Pennington et al., 2017; Xiao et al., 2018). Related approaches further demonstrate that explicitly regulating dominant singular components, such as through spectral normalization, stabilizes training by controlling the effective Lipschitz constants of learned transformations (Miyato et al., 2018; Chen et al., 2023b; Shi et al., 2023).

In MoE models, orthogonality- and decoupling-based methods reduce cross-expert redundancy and gradient interference, improving specialization and optimization (Liu et al., 2024). Building on this insight, our spectral decomposition separates shared low-rank components from expert-specific subspaces, removes common spectral "spikes," yields more benign conditioning across experts, and enables larger stable learning rates in practice.

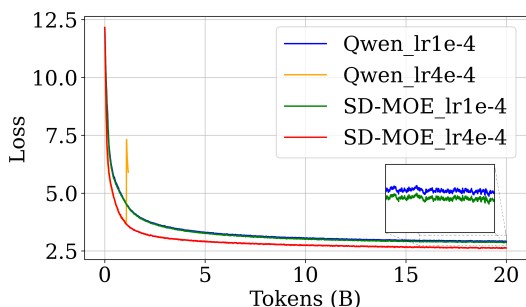

*Figure 8.* Compared to Qwen baseline, SD-MoE achieves lower loss and tolerates higher learning rates at 4×.

## 4.3. Analysis

**Expert & Gating Specialization.** To verify whether our model achieves effective specialization across experts as expected, we compute the principal subspace similarity among all experts as defined in Section 2. As shown in Figure 9, the pairwise principal subspace similarities are lower than 0.1 at convergence, indicating minimal overlap in their parameter directions. The gating mechanism also exhibits improved specialization. Compared to Qwen model in Figure 10(a), the row vectors of the gating matrix in SD-MoE in Figure 10(b) no longer exhibit strong alignment with the top singular directions of the expert parameters. Instead, they distribute their influence across a broader set of singular vectors, reflecting the use of richer, more diverse semantic cues during routing. We further ablate the gradient decomposition component in Appendix H.2. Removing gradient projection eliminates the gain, showing that parameter decomposition alone is insufficient to preserve expert specialization during training. This confirms that SD-MoE successfully promotes expert specialization and reduces parameter redundancy.

**Sensitivity to Common Rank.** We systematically evaluate $k \in \{2, 4, 8, 16, 32\}$. The average task performance is listed in Table 2. Despite the peak at $k = 8$, performance remains stable across the entire range, varying by less than 0.6%. This suggests that the shared structure present in human-generated text is inherently low-rank. Moreover, our method is robust to the choice of $k$: as long as $k$ is modestly sized, it effectively captures the common subspace and thereby enhances expert specialization.

| Metric | Small (2B–0.8B) | | | | Large (7B–1.5B) | |
|---|---|---|---|---|---|---|
| | DeepSeek | DeepSeek w. SD | Qwen | Qwen w. SD | Qwen | Qwen w. SD |
| ARC-challenge | 30.80 | **31.66** | 31.06 | **36.09** | 35.75 | **40.70** |
| ARC-easy | 62.16 | **63.43** | 62.04 | **68.39** | 69.91 | **71.84** |
| HellaSwag | 54.11 | **55.57** | 55.34 | **59.48** | 61.12 | **66.13** |
| LAMBADA_OPENAI | 39.63 | **40.21** | 35.09 | **42.97** | **49.31** | 47.29 |
| PIQA | 71.98 | **72.80** | 72.31 | **74.16** | 74.81 | **76.39** |
| RACE | **50.51** | 50.06 | 48.54 | **50.61** | 50.65 | **51.48** |
| SIQA | 40.53 | **41.25** | 40.89 | **41.71** | 41.76 | **41.91** |
| Winogrande | 54.33 | **56.91** | 58.33 | 58.33 | 59.43 | **62.43** |
| Avg | 50.51 | **51.49** | 50.45 | **53.97** | 55.34 | **57.23** |

*Table 1.* Accuracy (%) on 8 downstream tasks. Our SD-MoE framework is applied to various MoE architectures, including Qwen and DeepSeek, demonstrating its broad applicability across different model scales and structures. This table shows that integrating SD into existing MoE models improves performance across nearly all tasks, making it a reliable enhancement for any MoE-based architecture.

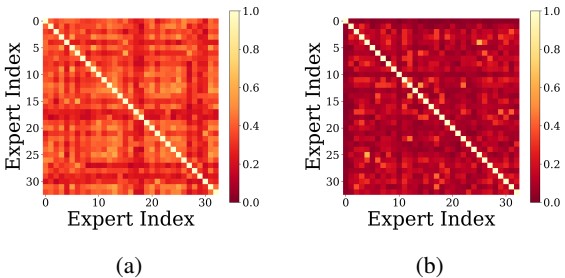

(a)  (b)

*Figure 9.* Pairwise principal subspace similarity between the dominant spectral subspaces of all experts in **(a)** Qwen baseline model, **(b)** our SD-MoE. The near-zero similarities across all expert pairs indicate effective decoupling of parameter directions, demonstrating successful expert specialization and reduced parameter redundancy. Supplementary results in more models and layers are in Appendix A Figure 22 and 23.

*Table 2.* Model performance with different common subspace rank $k$. Detailed results are in Appendix F Table 6.

| $k$ | 2 | 4 | 8 | 16 | 32 |
|---|---|---|---|---|---|
| Task Acc. | 53.64 | 53.39 | **53.97** | 53.57 | 53.68 |

**Computation Overhead.** Table 3 reports training throughput (tokens/sec) under identical hardware conditions. SD-MoE incurs around 5% overhead compared to standard MoE at training time, mainly from gradient projection; the periodic SVD refresh contributes less than 0.5%. At inference time, SD-MoE introduces no additional computational or communication cost beyond standard shared-expert MoE implementations, as the number of activated parameters remains unchanged. Overall, SD-MoE still achieves higher practical learning efficiency in wall-clock time by sustaining larger learning rates. A detailed efficiency breakdown is provided in Appendix H.

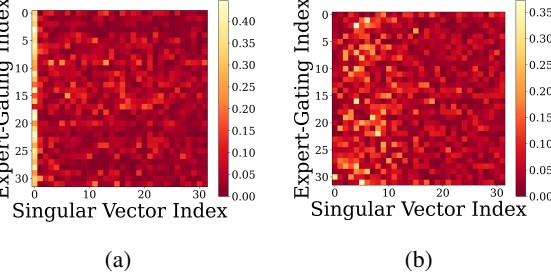

(a)  (b)

*Figure 10.* The alignment of row vectors of the gating matrix with the leading singular directions of the expert parameters in **(a)** Qwen baseline model and **(b)** our SD-MoE. In our SD-MoE, gating is not dominated by the largest singular vector. Supplementary results in more models and layers are in Appendix A Figure 20 and 21.

*Table 3.* Training throughput (tokens/second) under the same computational resources for each model scale.

| Model Scale | Qwen | SD-MoE (Ours) | Δ |
|---|---|---|---|
| 2B-A0.8B | 229K | 218K | 4.80% |
| 7B-A1.5B | 248K | 235K | 5.24% |

## 5. Related Works

**Mixture of Experts** (Shazeer et al., 2017) extended the MoE structure to deep neural networks and proposed a deep MoE model composed of multiple layers of routers and experts. Since then, the MoE layer with different base neural network structures (Dauphin et al., 2017; Vaswani et al., 2017) has been proposed. Following works explored various routing strategies like (i) letting tokens select the top-k experts (Lepikhin et al., 2021; Fedus et al., 2022; Zuo et al., 2022; Chi et al., 2022; Dai et al., 2022; Chen et al., 2023a), (ii) letting experts select the top-k tokens (Zhou et al., 2022), to (iii) globally decide expert assignment (Lewis et al., 2021;

Clark et al., 2022). Recently, MoEs have been widely adopted as a core component in large-scale language models deployed by leading organizations, including models such as Qwen (Yang et al., 2025), DeepSeek (Dai et al., 2024), and Mixtral (Jiang et al., 2024).

**Specialization failure and routing-based remedies.** Despite the success of sparse routing, MoE models do not always yield meaningful expert specialization in practice: experts may become functionally similar, some experts can dominate the routing decisions (Krishnamurthy et al., 2023; Cai et al., 2025), and the overall capacity gain can be reduced by redundancy and representation collapse (Chi et al., 2022; Liu et al., 2024; Zhang et al., 2025). Specialization from a semantic perspective have also been proposed (Dong et al., 2024; Zhou et al., 2025). A common line of work therefore focuses on *routing-level* improvements, such as load-balancing objectives, capacity constraints, and stochastic routing/regularization variants (Chen et al., 2023a; Zuo et al., 2022). More recently, alternative routing paradigms (e.g., expert-choice routing) have also been explored to mitigate imbalance by changing the assignment mechanism (Zhou et al., 2022).

## 6. Conclusion

This work addresses failure of expert specialization in MoE models. Our spectral analysis reveals that shared spectral components in parameters and gradients limit gating and expert differentiation. We resolve this with SD-MoE, which improves gating and expert specialization, enhances downstream performance, and is broadly applicable to existing MoE architectures.

## Acknowledgements

This research is supported in part by the National Natural Science Foundation of China under Grant 62090025.

## Impact Statement

This work aims to deliver insights to the Machine Learning community and will not lead to any direct societal consequences. While it is associated with LLMs that may output misleading or harmful content, such issues are outside the scope of this work and will not be particularly specified here.

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

## A. More Results in Analysis

Supplementary result for Section 2 are listed below in Figure 11 to Figure 18. All claims or observations we mentioned in Section 2 are consistent in all model layers.

## B. Derivation: a shared input subspace induces a shared gradient component

Recall $\mathbf{G}^{(i)} = \sum_{t \in \mathcal{B}_i} \boldsymbol{\delta}_t \mathbf{x}_t^\top$. Decompose each token input as $\mathbf{x}_t = \mathbf{P}_C \mathbf{x}_t + \mathbf{P}_{C^\perp} \mathbf{x}_t$, then

$$\mathbf{G}^{(i)} = \sum_{t \in \mathcal{B}_i} \boldsymbol{\delta}_t (\mathbf{P}_C \mathbf{x}_t)^\top + \sum_{t \in \mathcal{B}_i} \boldsymbol{\delta}_t (\mathbf{P}_{C^\perp} \mathbf{x}_t)^\top =: \mathbf{G}_C^{(i)} + \mathbf{R}^{(i)}. \tag{11}$$

Moreover, $\mathbf{G}_C^{(i)}$ has no right-action on $C^\perp$:

$$\mathbf{G}_C^{(i)} \mathbf{P}_{C^\perp} = \mathbf{0}, \qquad \text{equivalently} \qquad \mathbf{G}_C^{(i)} \mathbf{v} = \mathbf{0} \ \forall \mathbf{v} \in C^\perp, \tag{12}$$

since $\mathbf{P}_C \mathbf{v} = \mathbf{0}$ for all $\mathbf{v} \in C^\perp$ and thus $\mathbf{G}_C^{(i)} \mathbf{v} = \sum_{t \in \mathcal{B}_i} \boldsymbol{\delta}_t (\mathbf{P}_C \mathbf{x}_t)^\top \mathbf{v} = \sum_{t \in \mathcal{B}_i} \boldsymbol{\delta}_t \mathbf{x}_t^\top (\mathbf{P}_C \mathbf{v}) = \mathbf{0}$. Therefore, every expert gradient $\mathbf{G}^{(i)}$ contains a component $\mathbf{G}_C^{(i)}$ whose input-side (right-subspace) effect is supported on the shared subspace $C$.

## C. Method Details

Here we detail the initialization of unique experts in the orthogonal complement subspace, and the pseudo code is given in Algorithm 1. Given a weight matrix $\mathbf{W} \in \mathbb{R}^{m \times n}$ and its rank-$r$ shared component obtained via SVD, $\mathbf{W}_c = \mathbf{U}_k \boldsymbol{\Sigma}_k \mathbf{V}_k^\top$, where $\mathbf{U}_k \in \mathbb{R}^{m \times r}$ and $\mathbf{V}_k \in \mathbb{R}^{n \times r}$ have orthonormal columns, we aim to initialize each unique expert with a matrix supported entirely in the orthogonal complement of the shared subspace.

Specifically, for the row space, we seek a random matrix $\widetilde{\mathbf{U}} \in \mathbb{R}^{m \times (m-r)}$ such that:

1. $\widetilde{\mathbf{U}}^\top \widetilde{\mathbf{U}} = \mathbf{I}_{m-r}$ (columns are orthonormal);

2. $\mathbf{U}_k^\top \widetilde{\mathbf{U}} = \mathbf{0}$ (orthogonal to the shared row subspace).

An analogous condition holds for $\widetilde{\mathbf{V}} \in \mathbb{R}^{n \times (n-r)}$ in the column space.

To achieve this, we first sample a Gaussian random matrix $\mathbf{Z} \in \mathbb{R}^{m \times (m-r)}$. We then project it onto the orthogonal complement of $\mathrm{col}(\mathbf{U}_k)$ using the projector $\mathbf{P}_\perp = \mathbf{I} - \mathbf{U}_k \mathbf{U}_k^\top$:

$$\mathbf{Z}_\perp = (\mathbf{I} - \mathbf{U}_k \mathbf{U}_k^\top) \mathbf{Z}.$$

Since $\mathbf{Z}_\perp$ lies in the desired subspace but its columns are not necessarily orthogonal, we apply QR decomposition to obtain an orthonormal basis:

$$\widetilde{\mathbf{U}} = \mathrm{QR}(\mathbf{Z}_\perp).$$

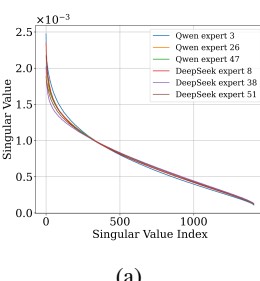 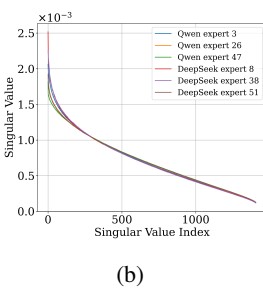 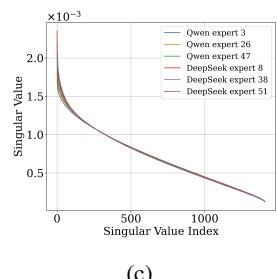 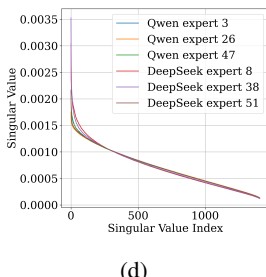

(a)  (b)  (c)  (d)

*Figure 11.* Expert singular value in Qwen (Qwen-Team, 2024) and DeepSeek (Dai et al., 2024) models from **(a)** layer 0, **(b)** layer 8, **(c)** layer 16, **(d)** layer 23. All experts in all model layers exhibit anisotropy, where 1% leading singular values constitute to more than 30% energy of the parameter spectrum.

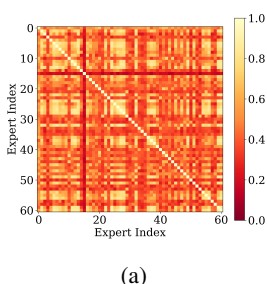 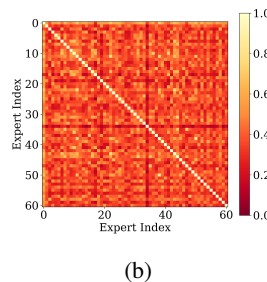 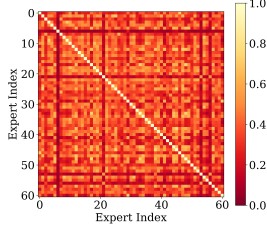 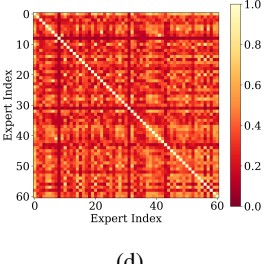

(a)  (b)  (c)  (d)

*Figure 12.* Pairwise expert principal similarity in Qwen (Qwen-Team, 2024) models from **(a)** layer 0, **(b)** layer 8, **(c)** layer 16, **(d)** layer 23. The overlap in expert dominant 1% spectral subspace is consistent in all model layers.

The same procedure is applied to the column space using $\mathbf{V}_k$. Finally, the unique weight is constructed as $\mathbf{W}_u = \widetilde{\mathbf{U}} \widetilde{\mathbf{\Sigma}} \widetilde{\mathbf{V}}^\top$, where $\widetilde{\mathbf{\Sigma}}$ is a diagonal matrix populated with the long-tail singular values of $\mathbf{W}$ (i.e., $\sigma_{r+1}, \ldots, \sigma_{\min(m,n)}$).

This procedure ensures that each unique expert starts in a direction fully orthogonal to the common subspace while maintaining proper scale and diversity across experts.

---

**Algorithm 1** Initialization of Unique Expert in Orthogonal Complement

---

**Require:** Shared bases $\mathbf{U}_k \in \mathbb{R}^{m \times r}, \mathbf{V}_k \in \mathbb{R}^{n \times r}$; number of experts $E$; long-tail singular values $\boldsymbol{\sigma}_{\text{tail}} \in \mathbb{R}^\ell$, $\ell = \min(m, n) - r$

**Ensure:** Unique weights $\{\mathbf{W}_u^{(i)}\}_{i=1}^E$

1: **for** $i = 1$ to $E$ **do**
2:     Sample $\mathbf{Z}_U \sim \mathcal{N}(0, 1)^{m \times (m-r)}$
3:     $\mathbf{Z}_U \leftarrow \mathbf{Z}_U - \mathbf{U}_k(\mathbf{U}_k^\top \mathbf{Z}_U)$ {Project to $\text{col}(\mathbf{U}_k)^\perp$}
4:     $\widetilde{\mathbf{U}}^{(i)} \leftarrow \text{QR}(\mathbf{Z}_U)$ {Ortho-normalize}
5:     Sample $\mathbf{Z}_V \sim \mathcal{N}(0, 1)^{n \times (n-r)}$
6:     $\mathbf{Z}_V \leftarrow \mathbf{Z}_V - \mathbf{V}_k(\mathbf{V}_k^\top \mathbf{Z}_V)$ {Project to $\text{col}(\mathbf{V}_k)^\perp$}
7:     $\widetilde{\mathbf{V}}^{(i)} \leftarrow \text{QR}(\mathbf{Z}_V)$
8:     Form $\widetilde{\mathbf{\Sigma}} = \text{diag}(\boldsymbol{\sigma}_{\text{tail}})$ {Pad with zeros if $m \neq n$}
9:     $\mathbf{W}_u^{(i)} \leftarrow \widetilde{\mathbf{U}}^{(i)} \widetilde{\mathbf{\Sigma}} (\widetilde{\mathbf{V}}^{(i)})^\top$
10: **end for**

---

## D. Detailed Model Configurations in Our Experiment

The following are the model configurations used in our experiments, where the keys follow the HuggingFace Transformers(Wolf et al., 2020) library.

*Table 4.* Model Configurations of Qwen Models

| Configs | 2B-A0.8B | 7B-A1.5B |
|---|---|---|
| num_hidden_layers | 32 | 40 |
| hidden_size | 1024 | 1536 |
| num_attention_heads | 16 | 16 |
| head_dim | 128 | 128 |
| num_key_value_heads | 8 | 8 |
| moe_intermediate_size | 2048 | 1024 |
| num_experts_per_tok | 2 | 4 |
| num_experts | 9(1 shared) | 33(1 shared) |

## E. Learning Rate Ablations on SD-MoE

As discussed in the main text, SD-MoE remains stable under learning rates more than $4\times$ larger than those tolerated by baseline MoE models. To further characterize the optimization behavior under high learning rates, we examine the evolution of the load-balance loss during training.

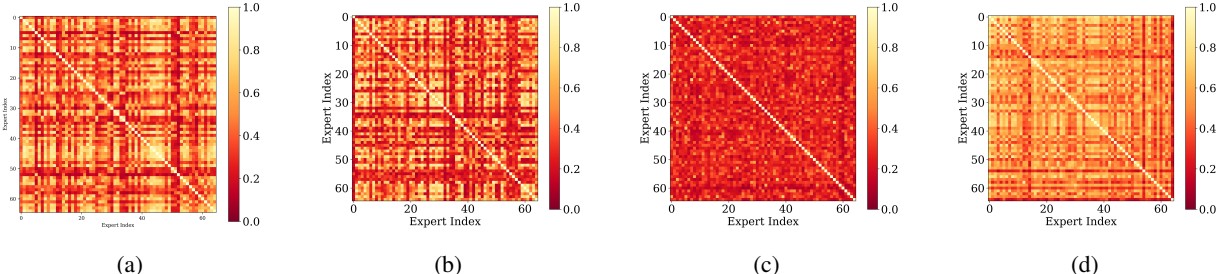

*Figure 13.* Pairwise expert principal similarity in DeepSeek (Dai et al., 2024) models from **(a)** layer 1, **(b)** layer 8, **(c)** layer 16, **(d)** layer 26. The overlap in expert dominant 1% spectral subspace is consistent in all model layers.

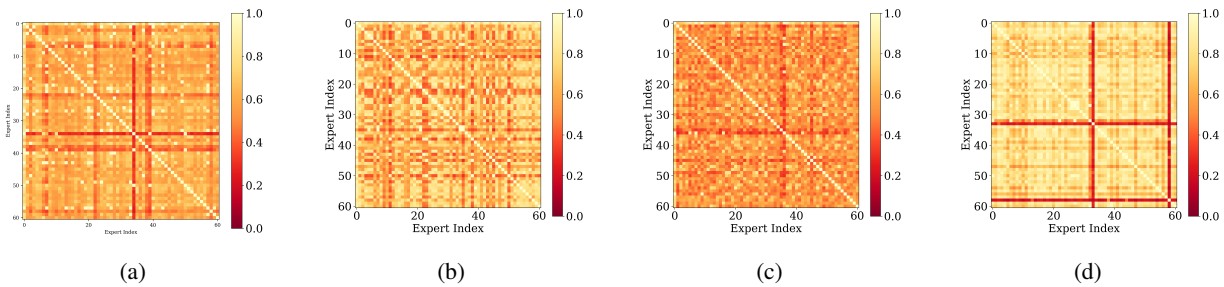

*Figure 14.* Pairwise expert gradient principal similarity in Qwen (Qwen-Team, 2024) models from **(a)** layer 0, **(b)** layer 8, **(c)** layer 16, **(d)** layer 23. The overlap in expert gradient dominant 1% spectral subspace is consistent in all model layers.

Specifically, we repeat the high–learning-rate experiments and track the load-balance loss at each training step until divergence occurs in the baseline models. As shown in Figure 24, divergence in the baseline MoE is consistently preceded by a sharp increase in load-balance loss, indicating severe expert imbalance and routing collapse. In contrast, SD-MoE maintains a low and stable load-balance loss throughout training, even under aggressive learning rates.

These results provide complementary evidence that SD-MoE improves routing stability under high learning rates, consistent with the more benign spectral conditioning induced by the proposed spectral decoupling.

# F. Common Subspace Rank Ablations on SD-MoE

We present ablation studies on the choice of the common subspace rank $k$ in SD-MoE. Table 6 summarizes the performance across different values of $k$. Notably, downstream task performance peaks at $k = 8$, suggesting that a rank-8 shared subspace may be sufficient to capture the dominant alignment structure across expert parameters under the selected model configuration.

*Table 5.* Configuration of Deepseek-2B-A0.8B

| Configs | Deepseek-2B-A0.8B |
| --- | --- |
| num_hidden_layers | 32 |
| hidden_size | 1024 |
| num_attention_heads | 16 |
| num_key_value_heads | 8 |
| kv_lora_rank | 64 |
| q_lora_rank | 192 |
| qk_nope_head_dim | 128 |
| qk_rope_head_dim | 64 |
| v_head_dim | 128 |
| moe_intermediate_size | 2048 |
| num_experts_per_tok | 2 |
| num_experts | 9(1 shared) |
| moe_layer_freq | 1 |
| first_k_dense_replace | 0 |
| n_group | 1 |
| topk_group | 1 |
| routed_scaling_factor | 1 |

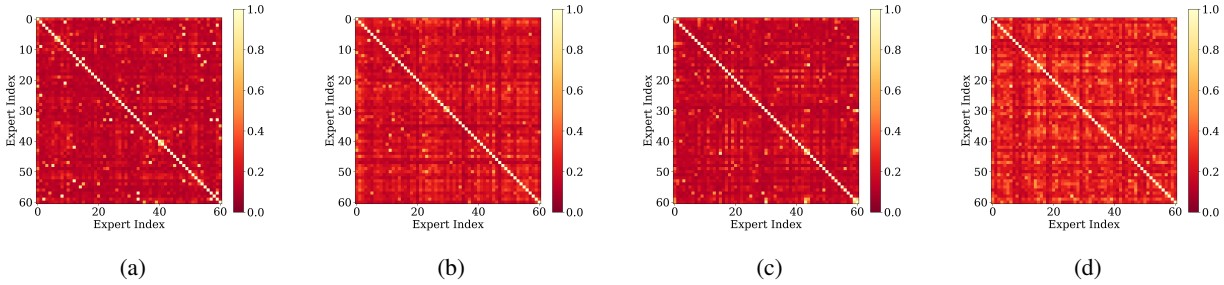

*Figure 15.* Pairwise expert gradient long-tail subspace principal similarity in Qwen (Qwen-Team, 2024) models from **(a)** layer 0, **(b)** layer 8, **(c)** layer 16, **(d)** layer 23. The overlap in expert gradient long-tail spectral subspace is consistent in all model layers.

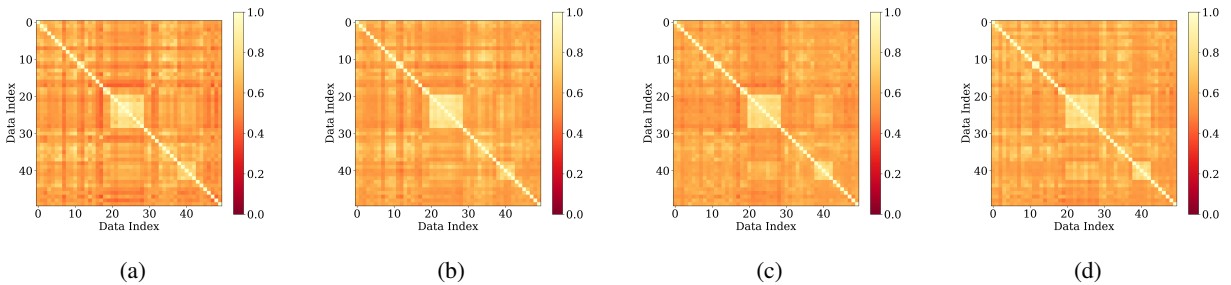

*Figure 16.* Data activation principal subspace principal similarity in Qwen (Qwen-Team, 2024) models from **(a)** layer 0, **(b)** layer 8, **(c)** layer 16, **(d)** layer 23. The overlap in data activation subspace is consistent in all model layers.

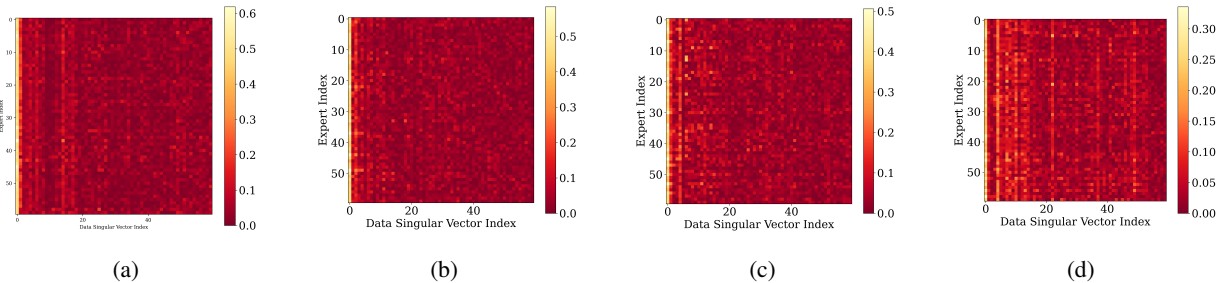

*Figure 17.* Gate vector and data activation alignment in Qwen (Qwen-Team, 2024) models from **(a)** layer 0, **(b)** layer 8, **(c)** layer 16, **(d)** layer 23. The gate vector is primarily aligned with the leading singular vectors of activation matrix in all model layers.

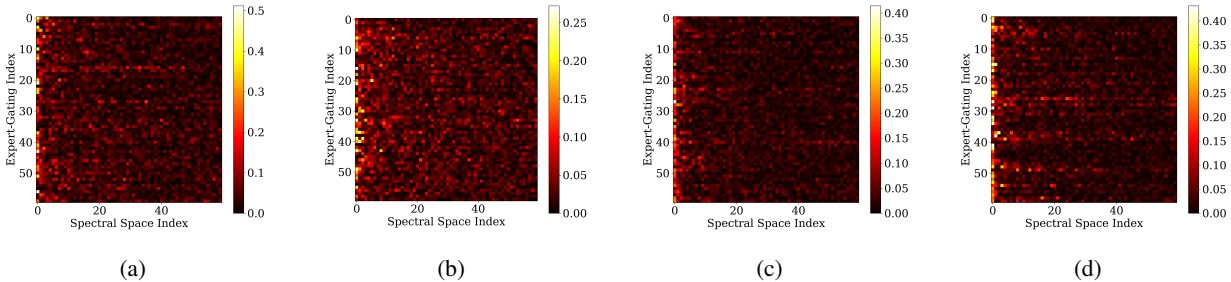

*Figure 18.* Gate vector and corresponding expert parameter singular vectors alignment in Qwen (Qwen-Team, 2024) models from **(a)** layer 0, **(b)** layer 8, **(c)** layer 16, **(d)** layer 23. The gate vector is primarily aligned with the leading singular vectors.

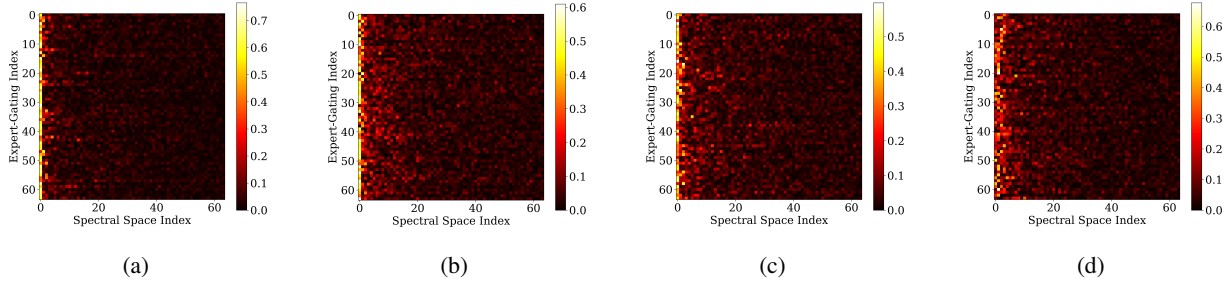

*Figure 19.* Gate vector and corresponding expert parameter singular vectors alignment in DeepSeek (Dai et al., 2024) models from **(a)** layer 1, **(b)** layer 9, **(c)** layer 17, **(d)** layer 25. The gate vector is primarily aligned with the leading singular vectors.

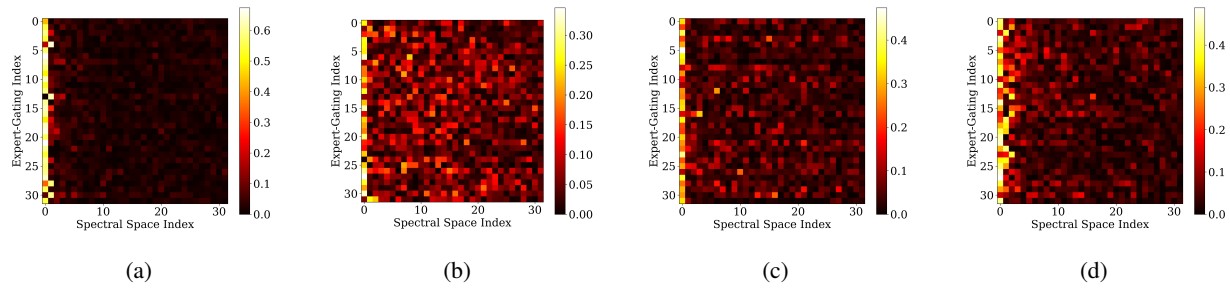

*Figure 20.* Gate vector and corresponding expert parameter singular vectors alignment in Qwen baseline models (pretrained from scratch in our experiments) from **(a)** layer 0, **(b)** layer 15, **(c)** layer 30, **(d)** layer 39. The gate vector is primarily aligned with the leading singular vectors.

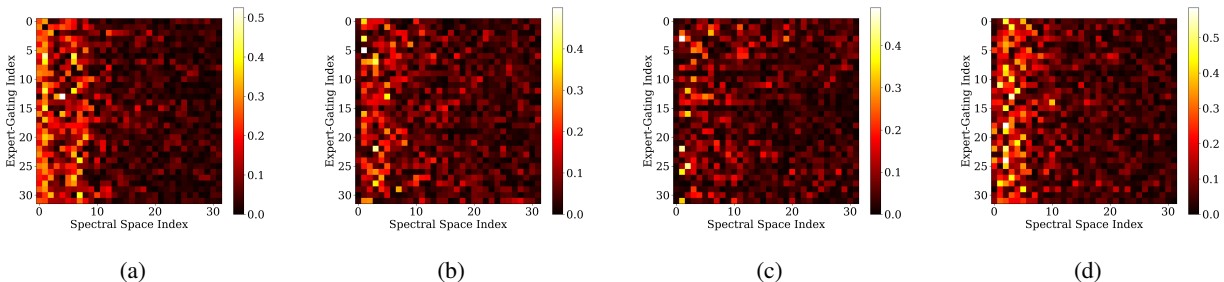

*Figure 21.* Gate vector and corresponding expert parameter singular vectors alignment in SD-MoE from **(a)** layer 0, **(b)** layer 15, **(c)** layer 30, **(d)** layer 39. Gating is not dominated by the largest singular vector.

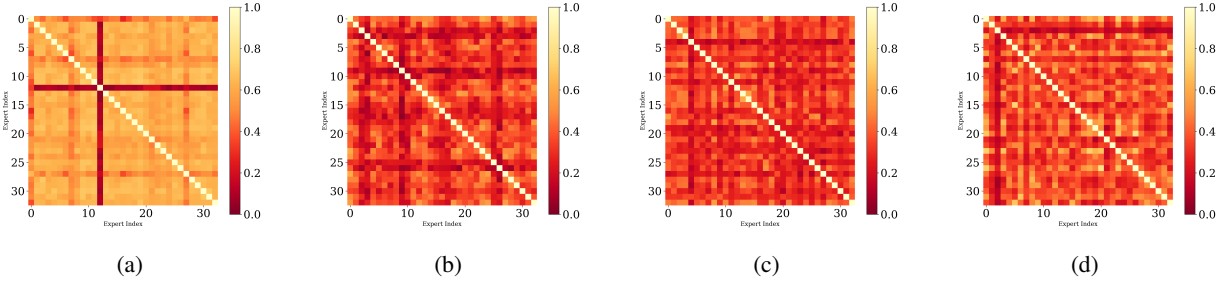

*Figure 22.* Pairwise expert principal similarity in Qwen baseline models (pretrained from scratch in our experiments) from **(a)** layer 1, **(b)** layer 15, **(c)** layer 30, **(d)** layer 39. The overlap in expert dominant 1% spectral subspace is consistent in all model layers.

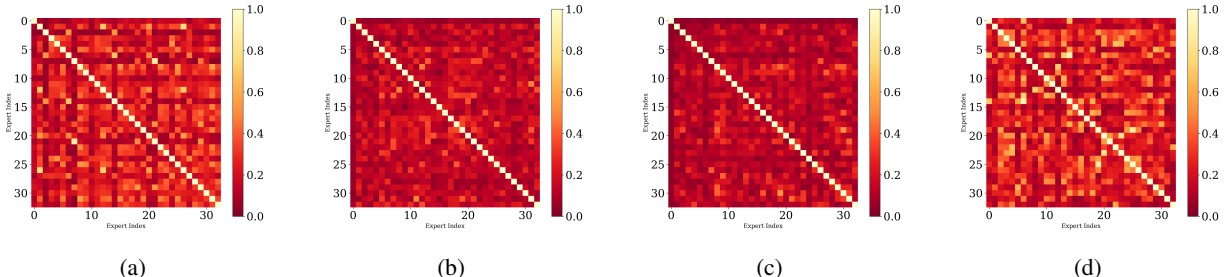

*Figure 23.* Pairwise expert principal similarity in SD-MoE from **(a)** layer 1, **(b)** layer 15, **(c)** layer 30, **(d)** layer 39. The near-zero similarities across all expert pairs indicate effective decoupling of parameter directions, demonstrating successful expert specialization and reduced parameter redundancy.

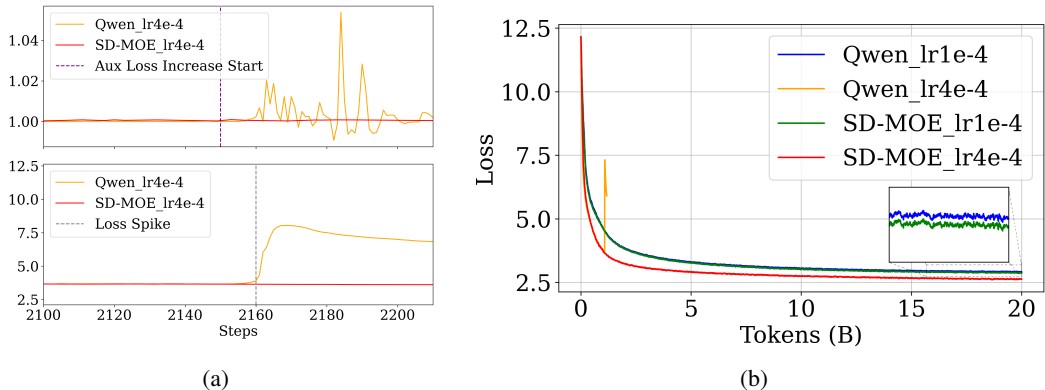

*Figure 24.* **(a)** Zoomed-in view around the loss spike: the top panel shows the auxiliary loss, and the bottom panel shows the training loss. A vertical line marks the onset of rising aux loss, which precedes the sharp spike in the main loss. **(b)** Full training loss curves.

| Metric | Rank=2 | Rank=4 | Rank=8 | Rank=16 | Rank=32 |
|---|---|---|---|---|---|
| ARC-challenge | 34.04 | 32.59 | **36.09** | 33.36 | 35.15 |
| ARC-easy | 66.04 | 65.99 | **68.39** | 67.34 | 67.51 |
| HellaSwag | 59.63 | **59.74** | 59.48 | 59.07 | 58.91 |
| LAMBADA_OPENAI | 42.79 | 42.56 | **42.97** | 42.32 | 42.77 |
| PIQA | **74.59** | 74.54 | 74.16 | 74.05 | 74.16 |
| RACE | 50.45 | 51.01 | 50.61 | 49.94 | **51.11** |
| SIQA | **41.86** | 41.30 | 41.71 | 41.04 | 41.45 |
| Winogrande | 59.75 | 59.43 | 58.33 | **61.40** | 58.41 |
| Avg | 53.64 | 53.39 | **53.97** | 53.57 | 53.68 |

*Table 6.* Accuracy (%) on 8 downstream tasks for Qwen-2B-A0.8B w. SD with varying common subspace ranks. The results show that a rank of 8 yields the best overall performance across all evaluated metrics, indicating that this might be an optimal choice for the shared subspace in our model configuration.

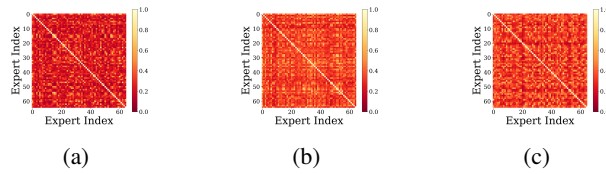

*Figure 25.* Dominant spectral alignment results on Moonlight-16B-A3B, a Muon-trained MoE model.

rameters. Therefore, SD-MoE does not introduce additional communication paths beyond standard shared-expert MoE implementations.

*Table 7.* Breakdown of the additional training overhead of SD-MoE. SD-MoE introduces no additional inference-time cost.

| Component | Overhead |
|---|---|
| Gradient projection | $\sim$4.5% |
| Periodic SVD refresh | $<$0.5% |
| Total training overhead | $\sim$5.0% |
| Inference overhead | 0% |

*Table 8.* Overhead of periodic SVD refresh under different update intervals. The cost remains small across all tested intervals.

| SVD refresh interval | SVD refresh overhead |
|---|---|
| Every 8 steps | 0.52% |
| Every 16 steps | 0.46% |
| Every 32 steps | 0.35% |

### H.1. Additional Evaluations

**Comparison with diversity-promoting baselines.** We compare SD-MoE with AES (Guo et al., 2026), a representative expert-specialization baseline based on diversity-promoting regularization. As shown in Table 9, SD-MoE outperforms AES by 2.39 average points under the same pre-training setting.

*Table 9.* Comparison with AES, a representative diversity-promoting expert-specialization baseline.

| Model | Average Task Accuracy |
|---|---|
| Baseline MoE | 50.45 |
| AES | 51.58 |
| SD-MoE | **53.97** |

## G. Dominant Spectral Alignment with a Muon Optimizer

We report the corresponding results on Moonlight-16B-A3B, a Muon-trained MoE model in Figure 25. The same dominant spectral alignment phenomenon is observed under this different optimizer.

## H. Detailed Efficiency and Communication Analysis

Table 7 breaks down the additional training overhead of SD-MoE, and Table 8 reports the overhead of different SVD refresh intervals. Our implementation is based on Megatron and uses the same parallelism configuration as the corresponding baseline MoE models: TP=2, PP=1, CP=1, and EP=2 on $4 \times 8$ GPUs. Both gradient projection and periodic SVD refresh are performed locally on expert pa-

**Generation evaluation on XSUM.** We further evaluate SD-MoE on XSUM summarization (Narayan et al., 2018) to test whether the gains transfer beyond multiple-choice benchmarks. As shown in Table 10, SD-MoE improves over the baseline on all ROUGE metrics.

**Training dynamics.** We track downstream performance throughout pre-training. Figure 26 shows that SD-MoE consistently outperforms the baseline across training tokens, indicating that the improvement is not limited to the final checkpoint.

**Effective capacity comparison.** We compare SD-MoE with a baseline MoE scaled to larger expert capacity. Figure 27 shows that the baseline requires approximately $1.25\times$

*Table 10.* XSUM summarization results. SD-MoE improves over the baseline across all ROUGE metrics.

| Method | ROUGE-1 | ROUGE-2 | ROUGE-L | Avg. |
|---|---|---|---|---|
| Baseline MoE | 33.10 | 12.08 | 26.51 | 23.90 |
| SD-MoE | **34.27** | **13.35** | **27.17** | **24.93** |

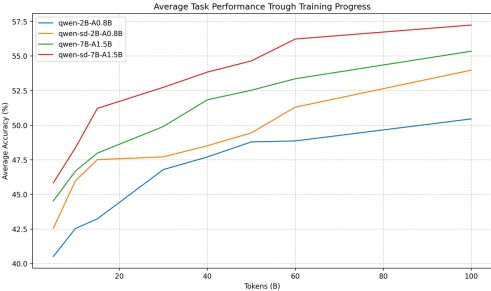

*Figure 26.* Average downstream performance throughout pre-training. Both the baseline and SD-MoE are trained from scratch on the same 100B-token DCLM subset. SD-MoE consistently outperforms the baseline across training tokens.

expert capacity to reach competitive performance with SD-MoE, suggesting that SD-MoE improves effective capacity utilization rather than merely increasing regularization strength.

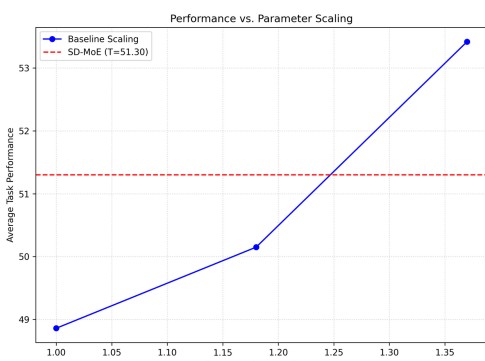

*Figure 27.* Effective capacity comparison. A baseline with approximately 1.25× expert capacity is required to reach competitive performance with SD-MoE.

## H.2. Ablation on Gradient Decomposition

Table 11 shows that parameter decomposition alone does not improve over the baseline, indicating that gradient decomposition is necessary.

*Table 11.* Ablation on gradient decomposition. Parameter decomposition alone does not improve over the baseline, indicating that gradient decomposition is necessary.

| Method | Average Performance |
|---|---|
| Baseline MoE | 48.86 |
| SD-MoE w/o Gradient Projection | 48.54 |

