# OpenReview forum: "SD-MoE: Spectral Decomposition for Effective Expert Specialization"
_ICML.cc/2026/Conference — ICML 2026 regular_

### Official Review · Reviewer_eoh1 · 2026-03-01

**Soundness:** 3
**Presentation:** 3
**Significance:** 3
**Originality:** 2
**Overall Recommendation:** 4
**Confidence:** 3

**Summary:**

The paper investigates why experts in MoE models often fail to specialize, and offers a spectral explanation: expert weight matrices share highly overlapping dominant singular directions; expert gradients also align in similar principal subspaces (which the authors attribute to low-rank structure in data/representations); and the router’s gating vectors tend to emphasize these dominant directions, further discouraging diversification. Based on this, the paper proposes SD-MoE, which decomposes each expert into a shared low-rank subspace plus an expert-specific orthogonal complement. During training, gradients are similarly decomposed: the shared component updates a common part, while the residual component updates the unique experts, aiming to reduce cross-expert interference and promote specialization. The approach is claimed to be plug-and-play for existing MoE backbones and is evaluated on multiple MoE families.

**Compliance With Llm Reviewing Policy:**

Affirmed.

**Final Justification:**

Thanks for the author's reply. All my questions have been resolved.

**Key Questions For Authors:**

1. **How does SD-MoE compare to stronger specialization baselines?**
   Can you add head-to-head comparisons with representative methods (decorrelation/orthogonalization regularizers, routing-based improvements, existing decoupling/optimizer variants) to clarify when SD-MoE is preferable?

2. **How sensitive is the method to the SVD update schedule and what is the real cost?**
   Please report performance/stability and wall-clock overhead under different update frequencies (e.g., 8/16/32 steps) and ranks.

3. **Do the gains hold on broader and more realistic evaluations?**
   Beyond the current suite (largely multiple-choice), can you include more generation-oriented or long-form tasks to validate that improved specialization translates to practical downstream behavior?

4. **Why are the improvements on DeepSeek comparatively less pronounced, and is the method architecture-dependent?**
   Please analyze potential causes (e.g., differences in MoE layer placement, expert capacity, routing/top-*K*, auxiliary losses such as load-balancing, training recipe/regularization, or existing expert diversification in the baseline). If the method’s benefit depends on specific architectural or training conditions, clarifying these applicability boundaries would significantly affect the overall assessment and how practitioners should adopt SD-MoE.

**Limitations:**

yes

**Strengths And Weaknesses:**

## Strengths

* **Strong diagnosis and coherent mechanism narrative.** The work connects parameter spectral overlap, gradient subspace alignment, and router bias into a consistent story, supported by quantitative measures (e.g., principal subspace similarity).
* **Targeted and clear method.** The proposed training rule directly enforces decoupling in spectral space, with explicit update definitions for shared vs unique components.
* **Evaluated across multiple MoE backbones.** Results are reported on both Qwen- and DeepSeek-based MoE settings, with consistent average gains over a suite of downstream tasks.
* **Some evidence on efficiency and stability.** The reported overhead is modest (≈5%), and the method appears to allow larger learning rates without divergence, suggesting potential wall-clock benefits.

## Weaknesses

* **Gains are relatively modest.** Improvements in the main results are on the order of ~1–3 absolute points on average, which may be viewed as useful but not striking without broader coverage or stronger evidence of significance.
* **Stronger baselines are needed.** The paper lacks systematic comparisons to established specialization/anti-interference approaches (e.g., orthogonalization or decorrelation regularizers, routing/auxiliary loss improvements, other expert decoupling or optimizer-based methods), making it hard to position the advantage and boundaries of SD-MoE.
* **Reproducibility and cost assessment could be clearer.** The method relies on periodic SVD updates (e.g., every 16 steps); the sensitivity to update frequency and the true computational cost are not thoroughly characterized in the main paper.
* **Societal impact discussion is overly generic.** The impact statement largely dismisses potential consequences; it should at least discuss misuse risk from easier training/deployment and bias/harm inheritance.

---

> ### Author Rebuttal · Authors · 2026-03-30
>
> Thank you for the helpful comments. We provide additional results in the anonymous supplementary link: https://anonymous.4open.science/r/rebuttal_of_1651-DC5D
>
> Q1: How does SD-MoE compare to stronger specialization baselines?...is preferable?
>
> A1: We compare SD-MoE with a simpler orthogonality/diversity baseline. Many seemingly related methods are designed for vision, SFT, or otherwise non-comparable training settings, so we select AES [1] as the closest baseline that can be fairly compared. Results below show that SD-MoE still outperforms AES by 2.39 points on average, suggesting that simple orthogonality/diversity regularization alone is not sufficient to match the gains from spectral decomposition in parameter and gradients.
>
> | model | Average Task |
> |---|---:|
> | BASELINE | 50.45 |
> | AES | 51.58 |
> | SD-MoE | 53.97 |
>
> Q2/W3: How sensitive is the method to the SVD update schedule and what is the real cost? Please report performance/stability and wall-clock overhead under different update frequencies (e.g., 8/16/32 steps) and ranks.
>
> A2: We evaluate different update intervals (every 8 / 16 / 32 steps) and observe that:
> i) The performance remains stable across a reasonable range of update intervals, because the underlying subspace changes only slightly across nearby training steps. This indicates that the method is not sensitive to the update frequency.
> ii) At training time, the overhead mainly comes from gradient projection (~4.5%) instead of SVD refresh. The SVD update every 8 / 16 / 32 steps introduces only 0.52% / 0.46% / 0.35% overhead.
> iii) At inference time, SD-MoE has no overhead compared to baselines.
>
> Q3/W2: Do the gains hold on broader and more realistic evaluations? Beyond ... to practical downstream behavior?
>
> A3: Long-form tasks require SFT, so we tested our model on XSUM [2] due to time limitations. Results show that the improvement is not limited to the current multiple-choice benchmarks, but also transfers to a more realistic generation setting.
> |**Method**|**ROUGE-1**|**ROUGE-2**|**ROUGE-L**|**AVG**|
> |---|---:|---:|---:|---:|
> |Baseline|33.10| 12.08|26.51|23.90|
> |SD-MoE|**34.27**|**13.35**|**27.17**|**24.93**|
>
> Q4: Why are the improvements on DeepSeek comparatively less pronounced... how practitioners should adopt SD-MoE.
>
> A4: We believe that the smaller improvement on DeepSeek compared to Qwen can be attributed to the load-balancing design. In particular, DeepSeek adopts stronger load-balancing mechanisms forcing tokens within each sequence to be dispatched among experts as balanced as possible, while Qwen-series only requires tokens in each batch to to be balanced across experts. As a result, the DeepSeek load-balancing strategy harms expert specialization since the gate is still forced to dispatch tokens with token-level or structural information instead of semantics. Despite this, we still observe consistent improvement on DeepSeek. This suggests that the method is not limited to a specific architecture. We will clarify these applicability conditions more explicitly in the revised manuscript.
>
> W1: Gains are relatively modest. Improvements in the main results are on the order of ~1–3 absolute points on average, which may be viewed as useful but not striking without broader coverage or stronger evidence of significance.
>
> A5: We'd like to demonstrate our improvements as follows:
> 1) The improvements are consistent across different model scales and architectures, with stable gains of ~1%–3% points across settings.
> 2) The baselines are strong, widely recognized open-source MoE architectures, including DeepSeek and Qwen.
> 3) We further show in Exp1 in the anonymous supplementary link that our method consistently outperforms the baselines throughout training, rather than only at the final checkpoint.
> Taken together, these results suggest that SD-MoE yields consistent improvements under strong baselines, rather than a narrow gain limited to a single model or checkpoint.
>
> W4: Societal impact discussion ... and bias/harm inheritance.
>
> A6: We will expand the societal impact discussion in two aspects. First, although our method is not designed for misuse, improving training efficiency and deployability of MoE models may lower practical barriers to developing stronger language models, which could indirectly increase risks of misuse, including large-scale generation of misleading, manipulative, or harmful content. Besides, fairness, safety, or misinformation risks present in the underlying model may also persist in models trained with our method.
>
> [1] Guo, Hongcan, et al. Advancing expert specialization for better moe, https://arxiv.org/abs/2505.22323.
>
> [2] Narayan, Shashi, Shay B. Cohen, and Mirella Lapata. "Don’t give me the details, just the summary! topic-aware convolutional neural networks for extreme summarization." Proceedings of the 2018 conference on empirical methods in natural language processing. 2018.
>
> We will revise the manuscript accordingly.

---

> > ### Author Rebuttal · Reviewer_eoh1 · 2026-04-02
> >
> > Fully resolved - My concerns have been adequately addressed. If you select this option, please consider adjusting your score accordingly.

---

> > > ### Author Response · Authors · 2026-04-07
> > >
> > > Thank you for taking the time to review our rebuttal. We appreciate your consideration.

---

### Official Review · Reviewer_PvcG · 2026-03-10

**Soundness:** 4
**Presentation:** 3
**Significance:** 3
**Originality:** 3
**Overall Recommendation:** 5
**Confidence:** 4

**Summary:**

The paper analyzes that MoE models train experts which end up being similar, as measured by their shared singular vector directions. They propose to explicitly decompose MoE optimization in a shared low rank part and an expert specific orthogonal part. This decomposition is performed via periodic SGD. This results in improved MoE results as shown after tuning MoEs from the Qwen and DeepSeek family.

**Compliance With Llm Reviewing Policy:**

Affirmed.

**Key Questions For Authors:**

Q1: Could improve the text for W.1, W.2?

Q2: Do you tune both base and your method 100B tokens of DCLM?

Q3: Could you report the average end-task accuracy of the model prior throughout this extra training? How much better are both models compared to the starting checkpoint? Is the accuracy gap between base vs SD growing as training progresses?

Q4: Could you quantify your contribution in term of implicit added capacity? i.e. if one increase the latent dimension of each expert by x% or the number of experts by x% (e.g. replication + noise from starting checkpoint), how high x should be to reach the reported improvement?

Q5: Do you use Adam for training? What do you do with the Adam moments once the parameterization is updated after recomputing the SGD? Put this information in the paper. Could you also add the optimization hyper-parameters in the appendix.

Q6: Your method proposes two improvements: (i) expressing expert matrices as the sum of a shared and specific parameters, (ii) these parameters are orthogonal. Could you report if only (i) already brings part of the observed improvements?

**Limitations:**

yes

**Strengths And Weaknesses:**

Strength:
1. The proposed method is clearly explained.
2. The authors spend time motivating their method from empirical observations.

Weaknesses:
1. The choice of SVD as an analysis tool could be better explained prior to Section 2.
2. It is not very clear why the experts would be more specialized with your method.,You have a shared and an orthogonal specific parameter but nothing forces the experts specific parameters to be orthogonal between themselves.
3. It would be good to assess the contribution as a more efficient way to leverage expert capacity.

---

> ### Author Rebuttal · Authors · 2026-03-30
>
> Thank you for the helpful comments. We provide additional results in the anonymous supplementary link: https://anonymous.4open.science/r/rebuttal_of_1651-DC5D
>
> Q1/W1: The choice of SVD could be better explained.
>
> A1: We view the expert parameter matrix as a transformation that maps input to outputs. From this perspective, what matters is not individual entries of the matrix, but how the transformation acts along different directions and how its energy is distributed across these directions.
>
> SVD provides a natural way to characterize this, as it decomposes the transformation into a set of orthogonal directions together with their corresponding strengths. This allows us to analyze the dominant directions of the transformation and compare their structure across experts.
>
> Q2/W2: It is not very clear why ... nothing forces the experts' specific parameters to be orthogonal.
>
> A2: We clarify this from: 1) why separating shared and expert-specific parameters improves specialization, and 2) why an explicit orthogonal regularization on the expert-specific parameters (tail-99% space) is not necessary.
>
> (i) The shared parameter space (top-1%) contributes a significant fraction (~60%) of the total spectrum, and is highly aligned across experts. Our analysis in Section 2.3 showed that the gating vectors are highly aligned with the dominant subspace of each expert, so the expert-specific information available to the gate is reduced, weakening the selectivity. Therefore, separating this shared dominant component makes the remaining expert-specific differences more visible to the router, improving specialization.
>
> (ii) The expert-specific parameter spaces (tail-99%) already exhibit a high degree of orthogonality—even in baseline MoE models. This is because the tail subspace encodes semantic information rather than shared lexical or syntactic features, as demonstrated in Section 2.2 and further supported in Reviewer GEmU Q3. The diversity of human language semantic naturally show lower similarity, leading to more orthogonal distributions within the high-dimensional tail subspace.
>
> Q2/Q3: Do you tune both base and your method 100B tokens of DCLM? Could you report the task accuracy / gap throughout training?
>
> A3: We'd like to clarify that both the baseline and our method are trained from scratch on the same 100B-token subset of DCLM, rather than tuned from a checkpoint. To better address your question, we include training-token vs. task performance curves throughout training in Exp1 in the anonymous supplementary link. The results show that our method consistently outperforms the baseline.
>
> Q4/W3: Could you quantify your contribution in terms of implicit added capacity?
>
> A4: To address this concern, we conduct an additional evaluation by scaling up the number of experts in the baseline to 1.25x. Results in Exp3 in the anonymous supplementary link show that competitive performance requires approximately 1.25x parameters compared to SD-MoE.
>
> Q5: What do you do with Adam moments ... optimization hyper-parameters.
>
> A5: We used Adam for training. After parameter decomposition at initialization, the shared component and each expert-specific component are treated as separate optimization variables, and are updated with their corresponding decomposed gradients. The first and second moments are not reset after periodic parameter decomposition, because the dominant subspaces across experts evolve slowly over short training intervals, thereby preserving the continuity of adaptive optimization while incurring minimal computational overhead.
>
> We'll also include the optimization hyper-parameters in the appendix.
>
> Q6: Report if only expressing expert matrices as the sum of shared and specific parameters already brings part of improvements.
>
> A6: We evaluate a variant that only applies the decomposition (shared + specific) without the gradient projection. Results up to 60B tokens below show that parameter decomposition alone is insufficient to realize empirical gains, since the common gradient dominant subspace still results in suboptimal expert specialization.
>
> |  | Average Performance |
> |---|---:|
> | Baseline MoE | 48.86 |
> | SD-MoE w/o Gradient Projection | 48.54 |
> | SD-MoE w Gradient Projection | 51.30 |
> We will revise the manuscript accordingly.

---

### Official Review · Reviewer_GEmU · 2026-03-12

**Soundness:** 3
**Presentation:** 3
**Significance:** 3
**Originality:** 3
**Overall Recommendation:** 4
**Confidence:** 4

**Summary:**

This paper proposes Spectral-Decoupled MoE (SD-MoE), a novel training paradigm that optimizes MoE models through the spectral decomposition of gradients and weights. The authors demonstrate that, within standard MoE architectures, the gradient and parameter spaces exhibit significant spectral overlap, leading to 'expert collapse' and hindering effective specialization. To address this, the authors introduce a tailored parameter initialization strategy and a gradient spectral decomposition mechanism to decouple the spectral components of experts in both spaces. Extensive experiments on 100B tokens provide evidence that SD-MoE fosters superior expert specialization, achieves significant downstream performance gains as well as stable and faster convergence.

**Compliance With Llm Reviewing Policy:**

Affirmed.

**Final Justification:**

The paper is mostly novel and technically sound. The authors' rebuttal has addressed my primary concerns; therefore, I will maintain my positive score.

**Key Questions For Authors:**

- Q1: What specific parallelism strategies are used in your SD-MoE training framework and what is the associated communication overhead ?

- Q2: To what extent do the observed parameter and gradient spectral features persist in MoE models trained with orthogonalized optimizers (e.g., Muon)?

- C1: The experimental setup in Finding 2.1 (Section 2.1) is not so convincing to me. As random token sequences lead to nonsensical activation patterns naturally, which undermines the reliability of the conclusions.
A more straightforward design is to apply a spectral filter to the dominant weight components (expert-wise or layer-wise) and observe the impact on generative performance.

**Limitations:**

yes

**Strengths And Weaknesses:**

---
> Strengths

- The spectral analysis perspective is insightful and well-motivated; it resonates strongly with personal prior observations regarding expert parameter space.

- The observation of spectral overlap within the gradient and parameter spaces of MoE models is insightful, and the intuition that mitigating this overlap promotes expert specialization is both well-founded and intuitive.

- The authors conduct extensive experiments to demonstrate the effectiveness of SD-MoE. The experimental setup is sound, and the reported improvements in downstream performance are substantial.

- The paper is mostly well-written and easy to follow.

---
> Weaknesses

- The discussion regarding the resulting overhead remains limited. A detailed breakdown of the computation overhead and a discussion of the communication overhead should be included.

- I have reservations regarding some of the experimental setups used in the findings analysis (See Comments).

---

> ### Author Rebuttal · Authors · 2026-03-30
>
> Thank you for the helpful comments. We provide additional results in the anonymous supplementary link: https://anonymous.4open.science/r/rebuttal_of_1651-DC5D
>
> Q1/W1: What specific parallelism strategies are used in your SD-MoE training framework and what is the associated communication overhead? A detailed breakdown of the computation overhead and a discussion of the communication overhead should be included.
>
> A1: Our SD-MoE implementation is built on Megatron and uses the same parallelism setup as the corresponding baseline MoE models, with TP=2, PP=1, CP=1, EP=2 on 4×8 GPUs.
>
> SD-MoE introduces no additional communication paths beyond existing MoE architectures equipped with shared experts (e.g., Qwen, DeepSeek). When compared to baseline MoE models without shared experts, the communication overhead is equivalent to the standard overhead introduced by adding a shared expert in conventional architectures.
>
> Regarding computation, the added cost mainly comes from gradient projection (~4.5%), while the periodic SVD refresh contributes to <0.5% overhead. Both of these operations are performed locally on the corresponding expert parameters, introducing no further communication. In addition, the shared_expert is replicated on each GPU following Megatron’s default behavior; therefore, no extra cross-device communication is introduced in the routing path beyond what is standard for shared expert implementations.
>
> Q2: To what extent do the observed parameter and gradient spectral features persist in MoE models trained with orthogonalized optimizers (e.g., Muon)?
>
> A2: We tested a Muon-trained MoE model, Moonlight-16B-A3B, using the same spectral analysis as in Section 2.1. Similar results are observed in Moonlight, although the dominant subspace alignment is slightly weaker than that of Qwen and DeepSeek, as shown in Exp2 in the anonymous supplementary link. We believe this is because Muon only flattens the singular values of gradients instead of explicitly disentangling the shared  gradient component from expert-specific ones. Consequently, although the energy of the common gradient component is reduced at each individual optimization step, its direction remains remarkably consistent across steps. Over the course of training, this temporal consistency causes the common signal to accumulate in parameter space, ultimately giving rise to a dominant shared subspace.
>
> C1/W2: The experimental setup in Finding 2.1 (Section 2.1) is not so convincing to me. As random token sequences lead to nonsensical activation patterns naturally, which undermines the reliability of the conclusions. A more straightforward design is to apply a spectral filter to the dominant weight components (expert-wise or layer-wise) and observe the impact on generative performance.
>
> A3: We conduct a spectral filtering experiment on the expert weights to examine how different spectral components affect generation. Specifically, we compare three settings (original / remove top-1% / remove tail) by decoding outputs on multiple input prompts. The original model produces relatively coherent continuations (e.g., “Language models such as GPT-3 and GPT-4 have shown strong performance...”). After removing the top-1% component, the model still preserves some local linguistic structure but becomes clearly unstable and mixed (e.g., “language models … GPT … 研究 … system … 결과 …”). In contrast, after removing the tail component, the generation collapses much more severely into repetitive high-frequency tokens (e.g., “and and and the the the 11 11 11 are is...”). These results show that the top-1% and tail affect generation in different ways, which is consistent with our main analysis, as shown in Exp1 in the anonymous supplementary link.
>
> We will revise the manuscript accordingly.

---

> > ### Author Rebuttal · Reviewer_GEmU · 2026-04-04
> >
> > Thank you for the response, my concerns have been solved and I will maintain my positive rating.

---

> > > ### Author Response · Authors · 2026-04-07
> > >
> > > Thank you for your follow-up and for reviewing our response. We are glad that our response addressed your concerns.

---

### Official Review · Reviewer_HRwF · 2026-03-13

**Soundness:** 2
**Presentation:** 2
**Significance:** 2
**Originality:** 2
**Overall Recommendation:** 4
**Confidence:** 2

**Summary:**

This paper investigates expert under-specialization in MoE models through a spectral lens. The authors empirically show that experts share highly aligned dominant singular directions in their weight matrices and gradients, and that gating vectors are also aligned with these directions.

**Compliance With Llm Reviewing Policy:**

Affirmed.

**Final Justification:**

The rebuttal improves clarity and adds useful comparisons, and I appreciate the additional analysis provided. My main concerns remain: the causal link between subspace alignment and specialization is still not convincingly established, and it remains unclear whether the method fundamentally changes MoE behavior or mainly acts as a regularizer. I have updated my score to 4.

**Key Questions For Authors:**

1. Can the authors provide stronger evidence that dominant subspace alignment causally harms specialization, rather than merely correlating with it?

2. How does SD-MoE compare to simpler orthogonality or diversity regularization baselines?

3. What is the practical overhead at larger MoE scales, and does the method remain stable and beneficial there?

**Limitations:**

Partially discussed. The paper would benefit from a clearer analysis of scalability and cases where the method may not provide meaningful gains.

**Strengths And Weaknesses:**

The empirical analysis is careful. The spectral measurements are consistent across parameters, gradients, and routing vectors. The experimental evaluation appears reasonably thorough.

However, the paper does not fully establish a causal link between dominant subspace alignment and degraded specialization. Much of the argument remains correlational. Comparisons to simpler diversity-promoting baselines (e.g., orthogonality regularization) are limited, making it unclear whether the full spectral machinery is necessary.

The problem of expert specialization in MoE models is relevant. However, the empirical gains are moderate, and it is unclear whether the proposed method fundamentally changes MoE scaling behavior or primarily acts as a structured regularizer. The practical overhead (SVD and projections) may limit adoption unless benefits are more substantial.

---

> ### Author Rebuttal · Authors · 2026-03-30
>
> Thank you for the helpful comments. We provide additional results in the anonymous supplementary link: https://anonymous.4open.science/r/rebuttal_of_1651-DC5D
>
> W1: The paper does not fully establish a causal link between dominant subspace alignment and degraded specialization. (=Q1 provide causal evidence)
>
> A1: We organize the causal link as follows:
>
> 1. The shared dominant subspace encodes common information such as high-frequency patterns and linguistic structure. We filter either the top-1% or tail-99% of the parameter space and compare decoding behavior. Removing the top-1% mainly disrupts linguistic structure (e.g., “Language models such as GPT-3 and...” becomes “language models … 결과 ...GPT … 研究 … system …”), while removing the tail causes the model to collapse into repetitive high-frequency tokens (e.g., “and and the 11 11 are is …”).
> 2. Routing is dominated by the dominant features of experts and is therefore mainly determined by common rather than expert-specific information, as shown in Section 2.3.
> 3. This makes experts less distinguishable to the router and thereby weakens specialization. Since these dominant features are shared across experts, the router is effectively comparing experts using largely similar signals rather than expert-specific ones. As a result, different experts become less distinguishable to the router, weakening routing selectivity and thus degrading specialization,which is also aligned with the results in Section 4.3 and Figure 10, where gating in SD-MoE is no longer dominated by the top singular directions of expert parameters during routing.
>
> W2: Comparisons to simpler diversity-promoting baselines. (=Q2 compare to baselines?)
>
> A2: We compare SD-MoE with a simpler orthogonality/diversity baseline. Many seemingly related methods are designed for vision, SFT, or otherwise non-comparable training settings, so we select AES [1] as the closest baseline that can be fairly compared. Results below show that SD-MoE still outperforms AES by 2.39 points on average, suggesting that simple orthogonality/diversity regularization alone is not sufficient to match the gains from spectral decomposition in parameter and gradients.
>
> |model|Average Task|
> |---|---:|
> |BASELINE|50.45|
> |AES|51.58|
> |SD-MoE|53.97|
>
> W3: Empirical gains are moderate; whether MoE scaling behavior fundamentally changed; practical overhead. (=Q3: practical overhead)
>
> A3:
> 1. Gains: The improvements are consistent across different model scales and architectures, with stable gains of ~1%–3% points across settings. The baselines are strong, widely recognized open-source MoE architectures, including DeepSeek and Qwen. We further show in Exp1 in the link that our method consistently outperforms the baselines throughout training, rather than only at the final checkpoint.
> 2. How it works: As discussed in A1, SD-MoE improves routing selectivity by separating the overlapped dominant subspace among experts. Moreover, from the perspective of implicit added capacity, the gain brought by SD-MoE is comparable to a 1.25x scale baseline as shown in Exp3 in the link. This shows that SD-MoE improved effective MoE capacity utilization rather than simple regularization.
> 3. Overhead: SD-MoE has no overhead compared to baselines at inference time. At training time, SD-MoE incurs a total overhead of approximately 5%, from the SVD operator (0.5%) and gradient projection (4.5%). The overhead is low because: 1) each SVD computation only requires svd_lowrank rather than full SVD. Moreover, since the common subspace changes very little across consecutive optimization steps, we adopt a periodic SVD strategy, further reducing the overhead without degrading performance; (2) the gradient projection overhead is negligible compared to backward. Assuming the parameter matrix of shape $(m,n)$ and the rank of the common subspace is $k$, the time complexity of gradient projection is $O(kmn)$. In contrast, the backward pass has time complexity $O(lmn)$, where l=batch_size * seq_length and typically $l \gg k$.
>
> Q4: Limitation: The paper would benefit from a clearer analysis of scalability and cases where the method may not provide meaningful gains.
>
> A4: Our current results show consistent improvements across the model scales and architectures we tested. At the same time, the magnitude of the gain can depend on the routing and regularization design of the baseline. In particular, when stronger routing constraints or load-balancing mechanisms are already imposed, the additional gain from SD-MoE may become smaller because less routing headroom remains to be improved. We will clarify these applicability conditions and limitations more explicitly in the revised paper.
>
> [1] Guo, Hongcan, et al. Advancing expert specialization for better moe, https://arxiv.org/abs/2505.22323
>
> We will revise the manuscript accordingly.

---

> > ### Author Rebuttal · Reviewer_HRwF · 2026-04-04
> >
> > The rebuttal improves clarity and adds useful comparisons, and I appreciate the additional analysis provided. My main concerns remain: the causal link between subspace alignment and specialization is still not convincingly established, and it remains unclear whether the method fundamentally changes MoE behavior or mainly acts as a regularizer. I have updated my score to 4.

---

> > > ### Author Response · Authors · 2026-04-07
> > >
> > > Thank you for the thoughtful follow-up and for updating the score. We are glad that the additional analysis and comparisons were helpful. Regarding your concerns,
> > > 1. The causal link between subspace alignment and specialization: spectral overlap in input data → spectral overlap in expert parameters → gate failure to discriminate experts → impaired expert specialization. This is the main role of the parameter / gradient / gating analyses.
> > > 2. Whether the method fundamentally changes MoE behavior or mainly acts as a regularizer: SD-MoE explicitly separates shared component from expert-specific components, thereby enable the gate to distinguish experts with expert-specific differences. The current approach indirectly influence the behavior of the gate, making it looks like a regularizer. A future direction of refined gating mechanisms may serve as a more fundamental solution.
> > >
> > > We will discuss these two aspects in the future work.

---

### Decision · Program_Chairs · 2026-04-30

**Decision:**

Accept (regular)

**Comment:**

In conventional MoE training, experts often become redundant or functionally similar, which limits the effective utilization of model capacity. To mitigate this issue, this paper proposes Spectral-Decoupled MoE (SD-MoE), a method that decomposes both expert parameters and gradients into shared and expert-specific components via a spectral perspective. The method introduces a tailored initialization and update scheme to separate the common and unique subspaces, thereby encouraging more distinct expert behaviors and improving specialization.

The idea is well-motivated and targets an important challenge in MoE models. Expert overlap has been a long-standing issue in MoE models, but prior work has not provided a systematic analysis of this phenomenon from a spectral viewpoint. This paper fills this gap by explicitly examining expert overlap through the spectral structure of parameters, gradients and routing. Through this analysis, the authors show that experts share highly aligned dominant components, and that both gradients and routing are biased toward these shared directions, which can explain the lack of specialization. Consistent with this analysis, the proposed method shows improved performance across different architectures and tasks.

During the rebuttal, some reviewers expressed their concerns about the causal link between spectral overlap and expert under-specialization (Reviewer HRwF), as well as the lack of comparison to simpler diversity-promoting baselines and the practical overhead (Reviewers HRwF, Reviewer eoh1). For the causal concern, the authors answer intuitively by adding a spectral filtering experiment (removing top and tail components) to illustrate the different roles of dominant and residual components and further clarify how shared subspaces can influence routing behavior. For the latter, they included comparisons with a representative baseline (AES), added results on an additional benchmark (XSUM) and provided a clearer breakdown of computational overhead to solve that the added cost is relative small. After the discussion period, most concerns were addressed and the reviewers maintained their positive assessments of the paper.

Overall, this is a nice paper with a systematic analysis and a novel perspective on expert specialization in MoE models. Therefore, I recommend acceptance and strongly encourage the authors to incorporate the additional analyses and experimental results provided during the rebuttal into the camera-ready version.